# Charge fluctuation entropy of Hawking radiation: a replica-free way to find large entropy

Alexey Milekhin and Amirhossein Tajdini

*Department of Physics, University of California, Santa Barbara, CA 93106, USA*

`milekhin@ucsb.edu, ahtajdini@ucsb.edu`

**Abstract**

We study the fluctuation entropy for two-dimensional matter systems with an internal symmetry coupled to Jackiw–Teitelboim(JT) gravity joined to a Minkowski region. The fluctuation entropy is the Shannon entropy associated to probabilities of finding particular charge for a region. We first consider a case where the matter has a global symmetry. We find that the fluctuation entropy of Hawking radiation shows an unbounded growth and exceeds the entanglement entropy in presence of islands. This indicates that the global symmetry is violated. We then discuss the fluctuation entropy for matter coupled to a two-dimensional gauge field. We find a lower bound on the gauge coupling $g_0$ in order to avoid a similar issue. Also, we point out a few puzzles related to the island prescription in presence of a gauge symmetry.

# 1  Introduction

A key aspect of quantum systems is entanglement between subsystems. Entanglement entropies and Renyi entropies are well-known quantities characterizing the entanglement between reduced density matrices in QFTs and quantum gravity. In the context of the black hole evaporation process, the entanglement entropy of Hawking radiation is a useful measure of information loss which has been a subject of the intense recent study [1–4].

If the theory has an internal symmetry, there are refined entanglement measures associated with the symmetry decomposition of reduced density matrices. Recently in the quantum many-body literature, there has been a growing interest in understanding the entanglement structure of charged sectors. In particular, the entanglement entropy in a charged sector and the Shannon entropy associated to charge probabilities are computed in various models [5–15] and even measured [16]. Non-Abelian symmetries were previously discussed in [17,18]. One of the goals of this paper is to review these results, which are mainly understood for Abelian symmetries, and investigate them further in the non-Abelian case.

Another goal of this paper is to study the interplay between global/gauge symmetries and dynamical gravity. Violation of global symmetries through wormholes has been discussed in the literature before [19–22]. Our work is similar to recent papers [23,24] where the violation of global symmetries by islands has been quantified. In [24], it was suggested that the global symmetries emerge from averaging for a bulk theory which is dual to an ensemble of boundary theories (see also [25] for a related observation in the context of the Narain duality [26,27]). In [23, 24], by explicitly computing various non-singlet correlation functions it was shown that the existence of islands leads to the violation of global symmetries. These non-singlet correlation functions vanish if the symmetry is gauged. Here we calculate another observable, the (charge) fluctuation entropy $S_f$ which does not have to vanish if the symmetry is gauged. What is more, we find that $S_f$ has an unbounded growth, which is not stopped even by the islands.

## 1.1  Summary

The logic of this paper is the following. Entanglement entropy is a very complicated quantity which is difficult to compute even for free theories. Its path integral evaluation commonly requires introduction of replicas and then taking a subtle $n \to 1$ limit. In the presence of dynamical gravity the story becomes even more complicated because of possible replica wormholes [3, 4]. However, in this paper we exploit a bound on entanglement entropy by studying very simple observables, which do not require replicas. Namely we study a charge probability distribution $p(q)$ in a certain region $R$. Naturally, its entropy $S_f$ (usually referred

to as fluctuation entropy in the literature) bounds the full entanglement entropy from below:

$$S_R \geq S_f = -\sum_q p(q) \log p(q). \tag{1.1}$$

The calculation of $p(q)$(and $S_f$) can be done by studying the expectation value of $e^{i\alpha Q_R}$, where the operator $Q_R$ measures the charge inside the region $R$:

$$p(q) \propto \int d\alpha \ e^{-i\alpha q} \langle e^{i\alpha Q_R} \rangle. \tag{1.2}$$

Notice that this computation does not require replicas. It has been discussed in the literature before, that the expectation value of $e^{i\alpha Q_R}$ should not be sensitive to islands [23, 28, 29], because it can be split into a product of local operators. Hence we will exploit the fact that its expectation value can be reliably computed semiclassically. For free fermion CFTs the finite temperature computation of $\langle e^{i\alpha Q_R} \rangle$ is elementary and $p(q)$ is Gaussian:

$$p(q) \propto \exp\left( -\frac{\pi^2}{2\log\left(\frac{\beta}{\pi\epsilon_{\mathrm{uv}}} \sinh\frac{\pi l}{\beta}\right)} q^2 \right), \tag{1.3}$$

$S_f$ shows an unbounded growth for large interval lengths $l$:

$$S_f \sim \frac{c}{2} \log \frac{l}{\beta} \text{ for } l \gg \beta, \text{ or } S_f \sim \frac{c}{2} \log \frac{2}{\pi} \log \frac{l}{\epsilon_{\mathrm{uv}}} \text{ for } \beta = \infty \tag{1.4}$$

We will perform this computation in detail in Section 2. In contrast, the central result of the island prescription is that the entanglement entropy stops growing. Moreover, irrespective of the island prescription, in a black hole background it should not exceed the Bekenstein–Hawking entropy. Therefore, the assumption that we have a global symmetry leads to a paradox.

How do we resolve it if the symmetry is gauged? In 1+1 dimensions gauge fields do not have propagating degrees of freedom and the gauge coupling is massive. Hence, if all the lengths are smaller than the inverse gauge coupling the computation of $\langle e^{i\alpha Q_R} \rangle$ stays the same. However, at large coupling(when the interval length is large compared to the inverse gauge coupling) the fluctuation entropy stops growing. For an eternal black hole, the entanglement entropy of radiation, and hence its fluctuation entropy $S_f$ should not exceed the black hole generalized entropy $S_{\mathrm{gen\ BH}}$ in order to avoid a "bag of gold" paradox:

$$S_{\mathrm{gen\ BH}} \gtrsim S_f. \tag{1.5}$$

One of the main results of this paper is that in the island setup, this inequality imposes a parametric *lower bound* on the gauge coupling constant $g_0$:

$$g_0 \gtrsim \frac{1}{\epsilon_p} \exp\left(-\frac{\pi}{2} \exp\left(\frac{\mathcal{O}(1)S_0}{c}\right)\right). \tag{1.6}$$

where $S_0$ is the extremal entropy, $c$ is CFT central charge, $\epsilon_p$ is the gravitational UV length cutoff. All our bounds are parametric, and the coefficients of $\mathcal{O}(1)$ not necessary are the same in different inequalities. We will discuss the details, such as cut-off dependence, Abelian vs non-Abelian case, in the main text. Our main argument for this bound is contained in Section 4.2. If the coupling is too small, inequality $S_{\text{gen BH}} \gtrsim S_f$ can be violated. Also using a similar logic we find that the renormalized dilaton value also obeys a bound:

$$\phi_r \lesssim c\epsilon_p \exp\left(\frac{\mathcal{O}(1)S_0}{c}\right), \tag{1.7}$$

This bound is not tied to the presence of gauge symmetry and instead is a consequence of the island formula. We will find that in JT theories arising from SYK and higher-dimensional black holes these bounds indeed holds.

Lower bound on the gauge coupling in terms of the gravitational coupling is very similar in spirit to weak gravity conjecture in higher dimensions [21, 30, 31]. However, there are also certain differences, mainly because our matter CFT is massless. We will discuss this more in the Conclusion.

We will also study another gravitational setup where the islands appear. Namely the gravitationally prepared states with bra-ket wormholes [32]. In this case we do not have inequalities originating from Bekenstein–Hawking entropy. However, the inequality (1.1) still must hold. We will demonstrate how gauge fields holonomies *at any value of the coupling* increase the island answer for $S_R$ in order to accommodate for $S_f$. Hence in this case we do not extract any bound on the gauge coupling.

To reiterate, there are two natural inequalities: $S_R \geq S_f$(self-consistency) and $S_{\text{gen BH}} \gtrsim S_f$(no black hole information paradox). The former is automatically satisfied once we incorporate gauge holonomies inside the replica wormholes. As for the latter one, we propose that it is satisfied only if the gauge coupling is big enough.

This solution sounds very specific to two-dimensions, where all gauge theories(at non-zero coupling) are confining[1]. However, there is a number of reasons in favor of this resolution:

- We will demonstrate that it is enough to gauge the symmetry in the gravitational region

---

[1]Strictly speaking, in theories with dynamical matter it is better to discuss charge screening, rather than confinement, as the actual Wilson loop observable might not exhibit an area law [33].

only[2].

- The bounds (1.6), (1.7) hold for "bottom-up" models of JT, such as Sachdev–Ye–Kitaev(SYK) and near-extremal 4d magnetic black holes.

- In Appendix A we argue on very general grounds that interactions in CFT lower the fluctuation entropy $S_f$.

- Finally, in higher dimensions for non-rotating black holes Hawking radiation is dominated by low angular momentum modes. Hence one can expect to apply effective 2d description even in higher dimensions. This question requires further investigation and we leave it for future work.

As a concluding remark, we mention that $S_f$ grows quite slowly, namely logarithmically at finite temperature. Hence, one might worry that $S_f$ exceeds $S_{\text{gen BH}}$(and the paradox arises) only at very long separations, when one has to presumably include baby universes [34]. In Section 5.1 we argue that these small corrections are not able to decrease the fluctuation entropy significantly. As a final comment, we mention that there have been attempts in the literature [35, 36] to obtain a stronger version of inequality (1.1).

This paper is organized as follows. In Sections 2.1 and 2.2, we review the calculation of the fluctuation entropy and the symmetry resolved entropy in 2d CFT with a $U(1)$ global symmetry. In Section 2.3, we compute the non-Abelian symmetry-resolved entropy and the fluctuation entropy for Wess–Zumino–Witten(WZW) models which have not been considered in the literature before. In Section 3, we calculate the fluctuation entropy for the models where the gravity region is joined to flat external bath regions [2, 3]. Finally, in Section 4, we consider a gauge field coupled to free fermions. We analyzed this case for small gauge coupling in Section 4.1. After that in Section 4.3 we discuss Yang–Mills theory in two dimensions in more detail. We will also discuss some subtleties of computing the entanglement entropy for a gauge theory in Section 4.4. After that in Sections 4.5, 4.6 we discuss how the island rule might be affected by gauge holonomies. Finally, in Section 4.7 we show that the paradox is resolved for large gauge coupling. Section 5 is dedicated to further comments. Section 5.1 explores perturbative and non-perturbative corrections to fluctuation entropy. In Section 5.2 we discuss a possible ensemble explanation for large fluctuation entropy. In Section 5.3 we demonstrate that the proposed bound on gauge coupling holds in simple "physical" examples JT arising from SYK and 4d near-extremal black holes.

In Conclusion we summarize our finding and list open questions.

---

[2]To avoid possible complications with the matter boundary condition, we can also require that the gauge coupling is big in the gravitational region only.

**Note added:** at the final stages of this project, ref. [37] appeared, which also studies non-Abelian symmetry resolved entropy.

## 2 Symmetry resolved entropy in QFTs

### 2.1 Review of symmetry resolved entanglement entropy in QFT

In this section, we briefly review the definition and computation of the symmetry resolved and fluctuation entropy for QFTs in the absence of gravity. Let us consider a QFT in $d+1$ dimensions with a $U(1)$ global symmetry. The charge operator is defined as $Q = \int j_0$ where $j_0$ is the local charge density operator. For any subsystem $R$, there is a corresponding charge operator $Q_R = \int_R j_0$. In a fixed charge state, the density matrix of the system $\rho$ satisfies $[\rho, Q] = 0$. By taking the partial trace, it follows that $[\rho_R, Q_R] = 0$ where $\rho_R$ is a density matrix of the region $R$. As a result, $\rho_R$ is decomposed into block diagonal density matrices associated to each charge sector,

$$\rho_R = \sum_q p_R(q)\rho_R(q), \tag{2.1}$$

where $p_R(q)$ is the probability of finding the density matrix with a charge $q$, such that $\sum_q p_R(q) = 1$.

The symmetry resolved entropy is defined as

$$S_R(q) = -\text{Tr}\left[\rho_R(q) \log \rho_R(q)\right]. \tag{2.2}$$

Note that even when the quantum state is an eigenvalue of the charge operator, the symmetry resolved entropy is non-trivial due to charge fluctuations among subsystems. Using this definition, the entanglement entropy can be written as

$$S_R = -\text{Tr}\rho_R \log \rho_R = \sum_q p_R(q)S_R(q) - \sum_q p_R(q) \log p_R(q), \tag{2.3}$$

where the first term and the second term in (2.3) are called the configuration and fluctuation entropy [16], respectively. Note that both of them are positive, hence there is an obvious bound:

$$S_R \geq -\sum_q p_R(q) \log p_R(q). \tag{2.4}$$

This simple observation will play a crucial role in the paradox we will encounter later in the paper. The probability $p_R(q)$ associated to finding a subsystem with a particular charge can

be found *without replicas*, simply by computing the moments of $p(q)_R$:

$$p(q)_R = \frac{1}{2\pi} \int_{-\pi}^{\pi} d\alpha \ e^{-i\alpha q} \langle e^{i\alpha Q_R} \rangle. \tag{2.5}$$

The symmetry resolved Renyi entropies are defined as

$$S_{n,R}(q) \equiv \frac{1}{1-n} \log \text{Tr}(\rho_R(q)^n). \tag{2.6}$$

A nice feature of symmetry resolved entropy is that it may be computed using the replica trick. To see this, let us define the charged moments

$$Z_n(\alpha) \equiv \text{Tr}\left(\rho_R^n e^{i\alpha Q_R}\right). \tag{2.7}$$

Taking the Fourier transform of charged moments projects into a fixed charge sector[3],

$$\mathcal{Z}_n(q) = \text{Tr}(\rho_R^n \Pi_q) = \int_{-\pi}^{\pi} \frac{d\alpha}{2\pi} e^{-iq\alpha} Z_n(\alpha), \tag{2.8}$$

where $\Pi_q$ is the projection operator. Hence, the charged Renyi entropy is written as

$$S_{n,R}(q) = \frac{1}{1-n} \log\left(\frac{\mathcal{Z}_n(q)}{\mathcal{Z}_1(q)^n}\right), \tag{2.9}$$

where the denominator factor is due to the normalization of the density matrix. By taking $n \to 1$ limit, we find

$$\lim_{n \to 1} S_{n,R}(q) = S_R(q). \tag{2.10}$$

## 2.2 Massless Dirac fermion

Let us consider an example of a massless free Dirac fermion in two-dimensions to compute the charged moments explicitly. This theory has a $U(1)$ global symmetry and is a CFT with $c = 1$. Equation 2.7 has the interpretation of a Euclidean path integral with an extra insertion of an Aharonov–Bohm (AB) flux (or equivalently an imaginary chemical potential) into one of the replica sheets. In practice, instead of calculating equation 2.7 on a $n$-sheeted replica manifold $\widetilde{\mathcal{B}}_n$, it is convenient to introduce $n$ fields $\Psi_l, l = 1, \cdots, n$ on a single sheet $\mathcal{B}_n = \widetilde{\mathcal{B}}_n/\mathbb{Z}_n$. In the absence of any AB flux, a fermion is assumed to satisfy the anti-periodic boundary condition. However, with an AB flux in region $R$, the fields $\{\Psi_l\}$ satisfy the twisted

---

[3]If the gauge group is $\mathbb{R}$, the integral must be taken over $\alpha \in (-\infty, +\infty)$.

boundary conditions

$$\vec{\Psi} = \begin{pmatrix} \Psi_1 \\ \Psi_2 \\ \dots \\ \Psi_n \end{pmatrix}, \qquad \vec{\Psi} \to T_\alpha \vec{\Psi}, \qquad T_\alpha = \begin{pmatrix} 0 & e^{i\alpha} & & \\ & 0 & 1 & \\ & & & \ddots \\ (-1)^{n+1} & & & 0 \end{pmatrix}. \qquad (2.11)$$

Here in writing (2.11), the flux is inserted in the first replica sheet. Also, fields in the complement region $\overline{R}$ are trivially identified. The matrix $T_\alpha$ has the following eigenvalues

$$\lambda_{k,\alpha} = e^{i\alpha/n} e^{2\pi i k/n}, \qquad k = -\frac{n-1}{2}, \cdots, \frac{n-1}{2}, \qquad (2.12)$$

and one can check that eigenvalues are the same regardless of which sheet the AB flux is inserted.

When $\alpha = 0$, it is well-known that the path integral is evaluated by correlation functions of twist fields. For any 2d CFT, these twist operators are primary operators with fixed scaling dimension $\Delta = \frac{c}{12}(n - 1/n)$. For arbitrary $\alpha$, the path integral is again computed by product of twist operators $\mathcal{T}_{n,\alpha}, \widetilde{\mathcal{T}}_{n,\alpha}$ with a modified scaling dimension $\Delta_{n,\alpha}$. For instance, a single interval region $[u, v]$ in the vacuum state for a massless Dirac fermion has the following charged moments [7, 9]

$$Z_n(\alpha) = \langle \mathcal{T}_{n,\alpha} \widetilde{\mathcal{T}}_{n,\alpha} \rangle = c_{n,\alpha}(v - u)^{-2\Delta_{n,\alpha}}, \qquad \Delta_{n,\alpha} = -\frac{1}{12}\left(n - \frac{1}{n}\right) - \frac{1}{n}\left(\frac{\alpha}{2\pi}\right)^2, \qquad (2.13)$$

where $c_{n,\alpha}$s are non-universal theory-dependent constants.

A convenient way to derive (2.13) is using bosonization techniques. This method also can be generalized to multiple intervals [11, 38, 39]. Let us define the boson field $\phi$ is defined through $j_\mu = \frac{1}{2\pi} \epsilon_{\mu\nu} \partial^\nu \phi$. This maps the free fermion theory to a massless scalar field with the Lagrangian $\mathcal{L} = \frac{1}{8\pi} \partial_\mu \phi \partial^\mu \phi$. The charged operator and the flux operator in any interval $[u, v]$, is then written as

$$Q_{[u,v]} = \frac{1}{2\pi} \int_u^v \partial_x \phi \, dx = \frac{1}{2\pi} (\phi(v) - \phi(u)), \qquad (2.14)$$

and

$$e^{i\alpha Q_{[u,v]}} = \mathcal{V}_\alpha(v) \mathcal{V}_{-\alpha}(u), \qquad (2.15)$$

where $\mathcal{V}_\alpha(x) \equiv e^{i\frac{\alpha}{2\pi}\phi(x)}$ in (2.15) is a vertex operator. Therefore, the twisted boundary conditions for a region is enforced by inserting local operators at the end points of an interval.

For a free fermion, Renyis are computed by a change of basis from $\Psi_k$ to $\widetilde{\Psi}_k$ which diagonalizes the matrix $T_\alpha$. $\widetilde{\Psi}_k$ is a multi-valued field that satisfy $\widetilde{\Psi}_k \to \lambda_{k,\alpha}\widetilde{\Psi}_k$ across the interval. The change in boundary conditions due to the replica trick and additional flux operators is incorporated by coupling the currents to background gauge fields $\widetilde{A}_\mu^k$ where $k = 1, 2, \cdots n$. Following [38], these background gauge fields satisfy[4]

$$\epsilon^{\mu\nu}\partial_\mu \widetilde{A}_\nu^k = \left(\frac{2\pi k}{n} + \frac{\alpha}{n}\right)\left[\delta^2(x-u) - \delta^2(x-v)\right].\tag{2.16}$$

Therefore, the charged moments are given by

$$
\begin{aligned}
Z_n(\alpha) &= \prod_{k=-\frac{(n-1)}{2}}^{\frac{n-1}{2}} \left\langle \exp\left(i\int \widetilde{A}_\mu^k j_k^\mu\right)\right\rangle \\
&= \prod_{k=-\frac{(n-1)}{2}}^{\frac{n-1}{2}} \left\langle \exp\left(\frac{i}{2\pi}\int \widetilde{A}_\mu^k \epsilon^{\mu\nu}\partial_\nu\phi_k\right)\right\rangle \\
&= \prod_{k=-\frac{(n-1)}{2}}^{\frac{n-1}{2}} \left\langle \exp\left(-\frac{i}{2\pi}\left(\frac{2\pi k}{n} + \frac{\alpha}{n}\right)(\phi_k(u) - \phi_k(v))\right)\right\rangle.
\end{aligned}
\tag{2.17}
$$

Here the expectation values are evaluated in the scalar theory. Eq. (2.7) yields the same answer as (2.13) after the Wick contraction and summing over $k$ in the exponent. The integral over $\alpha$ now becomes

$$\mathcal{Z}_n(q) \simeq l^{-\frac{1}{6}(n-1/n)}\int_{-\pi}^{\pi} d\alpha \exp\left(-i\alpha q - \frac{2}{n}\left(\frac{\alpha}{2\pi}\right)^2 \log l\right).\tag{2.18}$$

Here $l$ is the length of the interval in the UV cutoff units. For large $l$ the integral is concentrated at $\alpha = 0$, making charge-dependence weak. This was dubbed "equipartition of

---

[4]There is an ambiguity in defining the background gauge field. One may shift $\frac{2\pi k+\alpha}{n} \to \frac{2\pi k+\alpha}{n} + 2\pi m$, where $m$ is an integer, which has the same effect on the fermion phase as the gauge field defined in (2.16). As a result, one has to sum over all $m$ in calculating the charged moments. However, in the limit of large intervals which is the case in this paper, the dominant contribution is given by $m = 0$ sector. [11, 40]

entanglement" [6]. At the leading order for large $l$, one finds by doing a Gaussian integral[5]:

$$\mathcal{Z}_n(q) \simeq l^{-\frac{1}{6}(n-1/n)} \sqrt{\frac{n\pi}{2\log(l)}} e^{-\frac{n\pi^2 q^2}{2\log l}}, \tag{2.19}$$

In the large $l$ limit, the charged Renyi and the symmetry resolved entropy are given by

$$S_n(q) = S_n - \frac{1}{2}\log\left(\frac{2}{\pi}\log l\right) + \mathcal{O}(l^0), \qquad S_R(q) = S_R - \frac{1}{2}\log\left(\frac{2}{\pi}\log l\right) + \mathcal{O}(l^0), \tag{2.20}$$

where $S_n, S_R$ for a single interval are $\frac{1}{6}(\frac{n+1}{n})\log l$, $\frac{1}{3}\log l$, respectively.

One may also find the probabilities of finding the subsystem in a fixed charge from calculating $Z_1(\alpha)$ by a two-point function of vertex operators. This follows from a simple fact that $Z_1 = \langle e^{i\alpha Q_{[u,v]}} \rangle$ computes the Fourier transform of the charge distribution. For a large interval $R$ with length $l$, this probability is approximately Gaussian

$$p_R(q) \simeq \mathcal{Z}_1(q)/Z_1(\alpha = 0) = \sqrt{\frac{\pi}{2\log l}} e^{-\frac{\pi^2 q^2}{2\log l}}. \tag{2.21}$$

Furthermore, the fluctuation entropy in the large interval size $l$ is

$$S_f = -\sum_q p_R(q) \log p_R(q) \simeq \frac{1}{2}\log\left(\frac{2}{\pi}\log l\right). \tag{2.22}$$

Using (2.22), one easily verifies that $S_R = \sum_q p_R(q) S_R(q) + S_f$ in the large size limit [6]. It is important to comment on UV divergences. It is true that in QFT each term in the decomposition (2.3) is infinite and we need to add counterterms. However, these counterterms are local(localized on a boundary of region $R$) and do not grow with the interval length. So strictly speaking the inequality (2.4) involving $S_R$ and $S_f$ should be understood parametrically: if $S_f$ contains a contribution which is growing with volume, then $S_R$ should also grow at least as fast.

Notice that the negative term in eq. (2.20) exactly coincides with the fluctuation entropy. This is not accidental: equipartition of entanglement says that $S_R(q)$ is $q$-independent. Using the decomposition (2.3) we see that

$$S_R(q) = S_R - S_f \tag{2.23}$$

---

[5]If the expectation value of charge $\langle Q_R \rangle$ in region $R$ is non-zero, (2.19) would be peaked around $\langle Q_R \rangle$. This value in general is non-universal [7, 10].

[6]For the convention used above for $j_\mu$ and the scalar field Lagrangian, the vacuum correlation functions are given by $\langle e^{-i\int f(x)\phi(x)} \rangle = e^{\int d^2x d^2y f(x) \log|x-y| f(y)}$.

In case of $c$ complex Dirac fermions and $U(1)^c$ symmetry, it is easy to see that the above answer gets multiplied by $c$:

$$S_{f,U(1)^c} = \frac{c}{2} \log \left( \frac{2}{\pi} \log l \right).$$  (2.24)

Now we turn to the non-Abelian symmetry case.

## 2.3    A soluble case of non-Abelian symmetries: WZW models

Unlike the Abelian case, non-Abelian symmetry resolved entropies are almost completely overlooked in the literature(ref. [5] contains a small discussion of $SU(2)$ case). So the results in this section are new. Decomposition (2.1) can be easily generalized for non-Abelian symmetries:

$$\rho_R = \sum_\lambda p_R(\lambda) \rho_R(\lambda) \otimes \frac{\mathbf{1}_\lambda}{\dim(\lambda)},$$  (2.25)

where $\lambda$ is a representation, $\dim(\lambda)$ is its dimension and $\mathbf{1}_\lambda$ is the identity operator in that representation [7]. This last factor reflects the fact that $\rho_R$ commutes with all symmetry group elements. One can introduce symmetry-resolved entropies $S_R(\lambda)$:

$$S_R(\lambda) = - \operatorname{Tr} \rho_R(\lambda) \log \rho_R(\lambda).$$  (2.26)

Then the total entropy has the following decomposition:

$$S_R = \sum_\lambda p_R(\lambda) S_R(\lambda) - \sum_\lambda p_R(\lambda) \log \left( \frac{p_R(\lambda)}{\dim(\lambda)} \right)$$  (2.27)

Before we proceed with a non-Abelian computation, let us rederive $U(1)$ answer in a slightly different language. It is well-known that entanglement entropy for a single interval in 2d CFT depends on central charge only. The final answer for symmetry resolved entanglement above involves only one additional component(apart from the central charge and the interval length): the dimension $\Delta_V$ of the $U(1)$ twist field $\mathcal{V}_\alpha$. For free fermions(or, more generally, Luttinger liquid) it can be evaluated by bosonization and the result is quadratic in the twist angle $\alpha$, eq. (2.38).

Name "twist field", perhaps is a misnomer: the operator performing a $U(1)$ twist is generically non-local:

$$"\mathcal{V}_\alpha(l)\mathcal{V}_\alpha(0)" = \exp \left( i\alpha \int_0^l dx \ j_0(x) \right)$$  (2.28)

_______________

[7]This decomposition slightly differs from ref. [37], since we have separated an explicit $\mathbf{1}_\lambda$ factor. Hence our definitions of fluctuation entropy differ by $\log \dim(\lambda)$ term.

where $j_\mu$ is $U(1)$ current. By sheer conformal invariance, two-point function of $j_\mu$ is fixed[8]:

$$\langle j_0(x) j_0(0) \rangle = \frac{2}{(2\pi)^2} \frac{k}{x^2}. \tag{2.29}$$

If CFT has a gravity dual, $j_\mu$ should be dual to $U(1)$ gauge field in the bulk [14]. More generally, if the currents admit a free-field representation, the answer for $\mathcal{V}_\alpha(l) \mathcal{V}_\alpha(0)$ is given by two-point function of $j_\mu$ and is quadratic in $\alpha$:

$$\langle \mathcal{V}_\alpha(l) \mathcal{V}_\alpha(0) \rangle = \exp\left( \frac{k}{(2\pi)^2} \left( \frac{\alpha}{2\pi} \right)^2 \int_0^l dx dx' \frac{1}{(x-x')^2} \right) = \left( \frac{l}{\epsilon_{\text{uv}}} \right)^{-2k\left( \frac{\alpha}{2\pi} \right)^2}, \tag{2.30}$$

where $\epsilon_{\text{uv}}$ is the UV cut-off. This implies that

$$\Delta_\mathcal{V} = \overline{\Delta_\mathcal{V}} = \frac{k}{2} \left( \frac{\alpha}{2\pi} \right)^2 \tag{2.31}$$

Hence, the answer is quadratic in $\alpha$, the same as for free-fields. We want to generalize this for non-Abelian symmetries.

A collection of free fermions can be bosonized in terms of Wess–Zumino–Witten(WZW) model. Fermionic charge currents are the basic variables in this model. For $U(N)$ level $k$ its central charge is[9]

$$c = 1 + \frac{k(N^2 - 1)}{N + k} \tag{2.32}$$

WZW currents can be "further" bosonized by a collection of scalar/ghost fields by Wakimoto representation [41, 42] which exist for any levels and simply-laced Lie-algebras. This representation is useful for computing various correlators. Although it is important that we will need to use only generators of Cartan subalgebra, as other generators have a complicated expression in terms of free fields. For simplicity, we will concentrate on $U(N)$ but the discussion below can be easily generalized to any simply-laced Lie algebra.

Similar to $U(1)$ case, we consider an interval $R$ of length $l$ and we want to study "charge" distribution in the vacuum by computing the following matrix element[10]:

$$p(\alpha, \lambda) = \dim(\lambda) \langle \text{Tr}_\lambda\, e^{i\alpha Q_R} \rangle, \tag{2.33}$$

---

[8]The normalization is chosen such that $k = 1$ corresponds to a free massless fermion. Note that for Abelian case $k$ is the Luttinger parameter rather than a level.

[9]For $k = 1$, $c = N$ and we have $N$ complex Dirac fermions(fundamental representation). For $k = N$ $c = N(N-1)/2 + (N-1)/2 + 1$ and we have $N(N-1)/2$ complex Dirac fermions, $(N-1)/2$ real Dirac fermions and one extra complex Dirac fermion, which is the adjoint representation of $U(N)$.

[10]Factor $\dim(\lambda)$ is needed to impose the normalization $\sum_\lambda p(\alpha, \lambda) = \delta(\alpha)$. It follows from the standard identity for characters: $\sum_\lambda \dim(\lambda) \chi_\lambda(\alpha) = \delta(\alpha)$

where $\lambda$ is a representation(which substitutes charge in the non-Abelian case), $\mathrm{Tr}_\lambda\, e^{i\alpha Q_R}$ is the corresponding character, and $\alpha$ belongs to Lie algebra. If we want to project on representation $R$, we just need to integrate over group elements $U(\alpha) = e^{i\alpha}$ weighted with the corresponding $U(N)$ character function $\chi_\lambda(\alpha)$:

$$p(\lambda) = \int dU(\alpha)\, p(\alpha, \lambda)\chi_\lambda(\alpha) \tag{2.34}$$

As usual, we can decompose $U$ into a diagonal part

$$D = \mathrm{diag}(e^{i\alpha_1}, \ldots, e^{i\alpha_N}), \tag{2.35}$$

and extra rotation $\Lambda$: $U = \Lambda^\dagger D \Lambda$. The path integral computing (2.33) is actually $\Lambda$-independent, as the theory is $U(N)$-invariant. In other words, we make the Wilson line diagonal(it belongs to Cartan subalgebra). The measure $dU$, after integrating out $\Lambda$, yields Haar measure:

$$p(\lambda) = \dim(\lambda) \int d\alpha\, \mu_{\mathrm{Haar}}(\alpha)\chi_\lambda(\alpha)\langle \mathrm{Tr}_\lambda \,\mathrm{diag}(e^{i\alpha_1 Q_{R,1}}, \ldots, e^{i\alpha_N Q_{R,N}})\rangle \tag{2.36}$$

For $U(N)$ Haar measure is just a Vandermonde determinant for $e^{i\alpha}$:

$$\mu_{\mathrm{Haar}}(\alpha) = \prod_{a<b} \left(e^{i\alpha_a} - e^{i\alpha_b}\right)^2 \tag{2.37}$$

Now we need to find the dimensions of the vertex operators performing twists $D$. In WZW models the currents $j_{\mu,a}(z)$ belonging to Cartan subalgebra are essentially dual to a simple free boson operator $\propto \partial\phi$ [41, 42]. Hence, we can use a generalization of eq. (2.30):

$$\Delta_a = \frac{k}{4}\left(\frac{\alpha_a}{2\pi}\right)^2 \tag{2.38}$$

where $k$ is WZW level. Parameter $\alpha$ should be periodically continued from $[-\pi, \pi]$ to the whole line. Hence up to a $\lambda$−independent normalization constant we get

$$p(\lambda) = \dim(\lambda) \int_{-\pi}^{\pi} d\alpha\, \mu_{\mathrm{Haar}}(\alpha)\chi_\lambda(\alpha) \exp\left(-k \log\left(\frac{l}{\epsilon_{\mathrm{uv}}}\right) \sum_{a=1}^{N} \left(\frac{\alpha_a}{2\pi}\right)^2\right) \tag{2.39}$$

We got a familiar matrix model integral.

In Appendix B we compute it in the regime

$$k \log(l/\epsilon_{\mathrm{uv}}) \gg N \tag{2.40}$$

with the following result[11] for the representation distribution $p(\lambda)$:

$$p(\lambda) = Z^{-1} \dim(\lambda)^2 \exp\left(-\frac{g}{2} C_2(\lambda)\right), \tag{2.41}$$

where $C_2(\lambda)$ is the quadratic Casimir of representation $\lambda$, eq. (B.13), $Z$ is a normalization constant and

$$g = \frac{2\pi^2}{k \log(l/\epsilon_{\mathrm{uv}})}. \tag{2.42}$$

Interestingly, it looks like a sphere partition function of 2d Yang–Mills(YM) theory. From this expression it is straightforward to compute the fluctuation entropy in the limit $N \gg 1$:

$$S_f = \frac{N^2}{4} \log\left(k \log l/\epsilon_{\mathrm{uv}}\right) + \mathcal{O}((\log l)^0) \tag{2.43}$$

Details can be found in Appendix B. Also using the results from there, one can easily show that the equipartition of entanglement does not hold anymore because of the extra $\log \dim(\lambda)$ terms. Now in the fixed representation sector, the entropy is

$$S_R(\lambda) = S_R - \frac{N^2}{2} \log\left(k \log l/\epsilon_{\mathrm{uv}}\right) + \log \dim(\lambda). \tag{2.44}$$

As we shown in Appendix B, distribution $p(\lambda)$ yields the following mean value:

$$\langle \log \dim(\lambda) \rangle = \frac{N^2}{4} \log\left(k \log l/\epsilon_{\mathrm{uv}}\right). \tag{2.45}$$

Hence the decomposition (2.27) holds.

## 3 The fluctuation entropy in JT gravity

In this section, we study JT gravity plus matter coupled to an external bath system [2, 3]. This is a toy model in which gravity only exists in a part of spacetime and in particular, there is no dynamical gravity in the bath region. The matter exists in both gravitational and non-gravitational regions, and satisfies a transparent boundary condition at the interface. The matter content we consider is a large $c$ free CFT(massless Dirac fermions). More precisely,

---

[11]This agrees with the recent result of [37] obtained by different means.

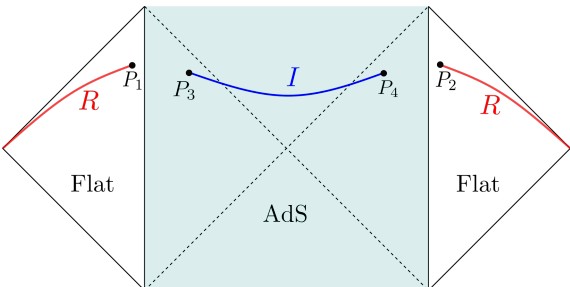

Figure 1: The eternal black hole in $AdS_2$ glued to the Minkowski space. The entanglement entropy of the radiation in region $R$ ceases to grow at late times due to existence of a non-trivial island $I$.

we consider the following theory (in Euclidean signature)

$$I = -\frac{S_0}{16\pi G_N}\left[\int_{\Sigma_2} R + 2\int_{\partial\Sigma_2} K\right] - \frac{1}{16\pi G_N}\left[\int_{\Sigma_2} \phi(R+2) + 2\phi_b \int_{\partial\Sigma_2} K\right] + I_{\text{CFT}}, \quad (3.1)$$

where $S_0$ is the extremal black hole entropy, and the boundary value of the dilaton is $\phi_b = \phi_r/\epsilon$. We also set $l_{\text{AdS}} = 1, 4G_N = 1$. The goal is to compute the fluctuation entropy and show that it is big. Large $c \gg 1$ is assumed in order to suppress quantum fluctuations in the gravitational path integral. We study two examples: the eternal black hole at a finite temperature and the boundary states in the context of bra-ket wormhole geometry. As mentioned in the last section, the fluctuation entropy is computed by $Z_1(\alpha)$ and in particular, it does not depend on the appearance of islands.

## 3.1 Eternal black hole coupled to a bath

This setup, depicted in figure 1, is considered in the context of the Page curve in [3]. The metric of eternal black hole, glued to flat space, is

$$ds_{\text{in}}^2 = \frac{4\pi^2}{\beta^2}\frac{dy d\overline{y}}{\sinh^2 \frac{\pi}{\beta}(y+\overline{y})}, \qquad ds_{\text{out}}^2 = \frac{1}{\epsilon^2}dy d\overline{y}, \tag{3.2}$$

$$y = \sigma + i\tau, \qquad \overline{y} = \sigma - i\tau, \qquad \tau \sim \tau + \beta. \tag{3.3}$$

The Lorentzian time is related to Euclidean time as $\tau = it$. The gravitational region is glued to the flat space at $\sigma = \epsilon$. We are interested in computing the fluctuation entropy of the region $R = [P_1, \infty_L) \cup [P_2, \infty_R)$. At late times, the entanglement entropy is dominated

by islands. The points in $(t, \sigma)$ have the following coordinates

$$P_1 = (i\beta/2 - t_b, b), \qquad P_2 = (t_b, b), \qquad P_3 = (i\beta/2 - t_a, -a), \qquad P_4 = (t_a, -a). \quad (3.4)$$

The entanglement entropy of region $R$ at late times is [3]

$$S(R) = 2S_0 + \frac{4\pi\phi_r}{\beta \tanh(2\pi a/\beta)} + \frac{c}{3} \log \left( \frac{\beta \left| \cosh\left(\frac{2\pi}{\beta}(a+b)\right) - \cosh\left(\frac{2\pi}{\beta}(t_a - t_b)\right) \right|}{\pi \epsilon_{UV,a} \epsilon_{UV,b} \sinh\left(\frac{2\pi}{\beta}a\right)} \right), \quad (3.5)$$

where $a, t_a$ are determined by the extremality condition $\partial_a S(R) = 0$,

$$t_a = t_b, \qquad \frac{\beta c}{12\pi\phi_r} \frac{\sinh(\frac{\pi}{\beta}(a-b))}{\sinh(\frac{\pi}{\beta}(a+b))} = \frac{1}{\sinh(2\pi a/\beta)}. \quad (3.6)$$

An important feature of the entanglement entropy is that it approaches a constant at late times in the eternal black hole case. More explicitly, if we work in the following regime

$$c \to \infty, \qquad S_0/c = \text{fixed and large}, \qquad \frac{\phi_r}{\beta c} \geq 1, \quad (3.7)$$

and $b \to 0$, we find [12]

$$S_R \approx 2S_0 + \frac{4\pi\phi_r}{\beta} + \frac{c}{3} \log \frac{\beta}{\epsilon_{UV,b}}, \quad (3.8)$$

where the first two terms in (3.8) coincide with the black hole coarse-grained entropy. The $\epsilon_{UV,a}$ is also absorbed in the definition of $S_0$.

In order to evaluate the fluctuation entropy, we first evaluate

$$Z_1(\alpha) \sim \langle \mathcal{V}_{-\alpha}(P_1) \mathcal{V}_\alpha(P_2) \rangle. \quad (3.9)$$

This correlation function is computed by mapping the points to a plane coordinate $x = e^{\frac{2\pi}{\beta}y}$ with metric $ds^2 = \Omega^{-2} dx d\bar{x}$ given by (3.2). The result for each $U(1)$ global symmetry is

$$Z_1(\alpha) \sim \exp\left[ -\left(\frac{\alpha}{2\pi}\right)^2 \log \frac{|x_{12}|^2}{\Omega_1 \Omega_2} \right] = \exp\left[ -2\left(\frac{\alpha}{2\pi}\right)^2 \log\left( \frac{\beta}{\pi \epsilon_{UV,b}} \cosh(\frac{2\pi t}{\beta}) \right) \right]. \quad (3.10)$$

---

[12] We approximated $\tanh(2\pi a/\beta) \approx 1$ for the range of parameters written in equation (3.7).

By taking a Fourier transform as (2.21), we find in the Abelian $U(1)^c$ [13] case

$$-\sum_q p_R(q) \log p_R(q) = \frac{c}{2} \log \log \left[ \frac{\beta}{\pi \epsilon_{UV,b}} \cosh \left( \frac{2\pi t}{\beta} \right) \right] + \mathcal{O}(1). \qquad (3.11)$$

Comparing with eq. (3.8), this shows that at large times of order $t \sim t^*$,

$$t^* = \frac{\beta}{2\pi} \exp \left( \frac{4S_0}{c} + \frac{8\pi \phi_r}{c\beta} + \frac{2}{3} \log \frac{\beta}{\epsilon_{UV,b}} \right), \qquad (3.12)$$

the fluctuation entropy of Hawking radiation becomes larger than the entanglement entropy of Hawking radiation and, more importantly, black hole coarse-grained entropy. Clearly this does not make sense. This indicates that the assumption about the underlying global symmetry can not be exact for all times and should be violated for $t \sim t^*$. In the next example, we will see that by considering large subsystems, one may reach to the same conclusion about the violation of global symmetries.

The reader might worry about the dependence of $t^*$ on UV cutoff in (3.12). It is believed that the generalized entropy is a finite quantity due to the cancellation between renormalization of $G_N$ and the cutoff dependence of entanglement entropy [43]. However, since there is no gravity in the bath, $\epsilon_{UV,b}$ cutoff can be arbitrarily small which renders $t^*$ to be arbitrarily large. We think this is an artifact of the models without gravity and in more realistic models, $\epsilon_{UV,b}$ at most is an order one number (in the Planck unit, i.e. $l_p = 1$). Note that the dependence on cutoff in exponent is only logarithmic compared to $S_0/c$ and in particular it is not enhanced by a factor of $c$. It is possible to remove the cutoff dependence completely by finding an appropriate definition for the mutual information in a given charge sector (see [44] for a partial success in that direction) and repeating the argument in section 3 of [45], however, we have not succeeded in doing that.

## 3.2 Bra-ket wormhole

Another interesting setup, introduced in [32], studied the island formula for gravitationally prepared states. The matter content of the theory is the same as the rest of this section. The semi-classical realization of the partition function $Z_1$ has two saddle points (a) and (b) in Figure 2. The saddle point (b) is a saddle where the bra and ket are connected by gravitational region. When the spatial length of the system is infinite, this is the dominant saddle point for the partition function and we can ignore the background geometry (a).

---

[13]In the non-Abelian $U(N)$ case and adjoint matter we should substitute $c/2 \to c/4$ and multiply cosh by level $k$.

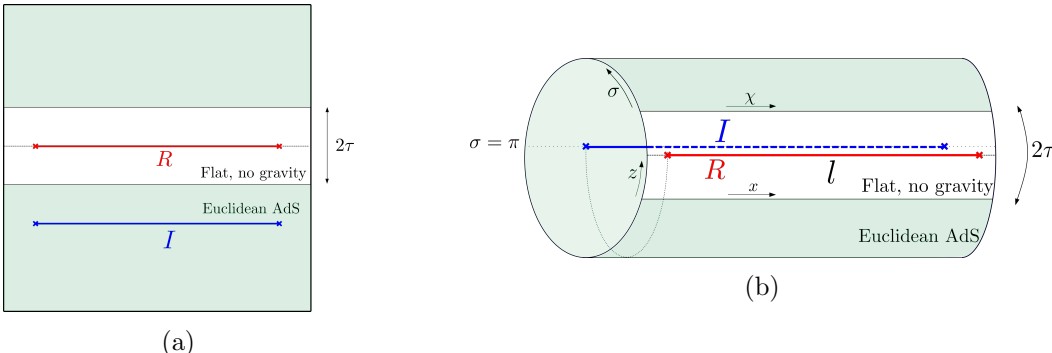

(a)

(b)

Figure 2: The bra-ket wormhole setup. The CFT state in the non-gravitational region(white) is prepared by doing the Euclidean path integral in $AdS_2$ where the gravity is dynamical(light green). CFT fields exist in both regions. (a) Computing CFT entanglement entropy for a single interval(solid blue) can include an island in the gravitational region(extra twist points connected to the interval by a dashed line). (b) Replica path integral can join bra and ket gravitational regions, forming a cylinder.

Following the notation in [32], we label the Euclidean time and spatial coordinates as $z, x$ in the non-gravitational region for the cylinder geometry in Figure 2 (b). For gravity, the coordinates are denoted by $\sigma, \chi$. The metric in different regions are

$$ds^2 = \frac{dz^2 + dx^2}{\epsilon^2}, \qquad 0 \le z \le 2\tau, \tag{3.13}$$

$$ds^2 = \frac{d\sigma^2 + d\chi^2}{\sin^2(\sigma)}, \qquad \sigma_c \le \sigma \le \pi - \sigma_c. \tag{3.14}$$

Solving the equation of motion after joining these two regions relate parameters as the following

$$\sigma_c = \frac{\pi c}{8\phi_r}\epsilon, \qquad \chi = \frac{\pi c}{8\phi_r}x. \tag{3.15}$$

The physical inverse temperature is given by $\beta_p = \frac{8\phi_r}{c}$. Considering a large interval $R$ in the non-gravitational with length $l$ and $z = \tau$, its entanglement entropy involves an island $I$ in the gravitational region as shown in figure 2. The island location is at $\sigma_* = \pi/2$. The generalized entropy for the region $R$ is

$$S_R^{\text{island}} = 2S_0 + \frac{c}{2} + \frac{c}{3}\log\left(\frac{\beta_p}{\pi\epsilon_{\text{uv}}}\right). \tag{3.16}$$

In order to find $Z_1(\alpha)$, we compute the two-point function of vertex operators at end points

of the interval in the non-gravitational region. The answer for each $U(1)$ is

$$\frac{Z_1(\alpha)}{Z_1(0)} = c_\alpha \exp\left[-2\left(\frac{\alpha}{2\pi}\right)^2 \log\left(\frac{\beta_p}{\pi\epsilon}\sinh \pi l/\beta_p\right)\right],\tag{3.17}$$

where $c_\alpha$ is a non-universal constant. Taking the large $l$ limit, and working in the same limits as (3.7), we find the fluctuation entropy for $U(1)^c$ as

$$S_f(R) \simeq \frac{c}{2}\log(\pi l/\beta_p),\tag{3.18}$$

which will be larger than $S_{\text{island}}(R)$ for

$$l \gtrsim l^* = \beta_p/\pi \exp\left[2S_{\text{island}}/c\right] \simeq \beta_p/\pi \exp\left(4S_0/c + \mathcal{O}(1)\right).\tag{3.19}$$

Similar to the Section 3.1, the main point is having an upper bound for the entanglement entropy, is not consistent with having a matter with global symmetries if the global symmetries are exact for lengths of order (3.19). Note that the island rule in the argument is used only to justify having a finite (and not growing) entanglement entropy for large regions.

## 4 Entanglement entropy plus gauge fields in 2d

### 4.1 Overview of difficulties

In the previous section, we discussed that when there is a bulk global symmetry, the fluctuation entropy of subsystems may indefinitely grow. We now discuss what happens if there is a gauge field coupled to massless Dirac fermions. In this section, we modify the toy model in section 2.2 by including a gauge field in the gravitational region only. A similar discrete toy model was considered in [46]. This has an advantage that one defines the entanglement entropy and fluctuation entropy for a region in the bath which by itself has a global symmetry.

First of all, once we couple a free Dirac fermion to YM theory, we no longer have a CFT. However, we would like to argue that the fluctuation entropy calculations in Section 3 stay essentially the same as long as the coupling is small. With fermions interacting with a dynamical gauge fields we need to take into account three things:

- Propagating gauge degrees of freedom.

- Fermions are no longer free.

- Extra possible holonomy degrees of freedom in the path integral.

- Change in black hole entropy.

In 2d we are lucky and the first two are simple. In 2d massless vector bosons do not have propagating degrees of freedom. Nonetheless gauge degrees of freedom can still contribute to the entanglement entropy [47, 48], but this contribution does not depend on interval length. Moreover gauge coupling $g_0$ is massive, hence as long it is smaller than all (inverse) lengths, we can simply ignore it[14].

The last two issues are more complicated. Again, the fluctuation entropy calculation does not require replicas so the space-time topology is trivial and this issue simply does not exist. Also it has been discussed in the literature before that the expectation value of $e^{i\alpha Q_R}$ should not be sensitive to islands [23, 28, 29]. The idea is that it performs a symmetry transformation on $R$ which can be split into smaller symmetry transformations acting locally on small subintervals of $R$, plus possible boundary terms. Local operators supported on tiny subintervals of $R$ should not be able to see the island. However, for the genuine entanglement entropy $S_{gauge}(R \cup I)$, the replicated manifold has a non-trivial topology and holonomies do play a major role. We postpone this discussion until Section 4.4.

Another issue is the change in black hole entropy due to non-trivial charge distribution. Since we are using Euclidean path integral for state preparation, all our computations are done at a fixed (namely zero) chemical potential. Abelian 2d YM theory on a disk can be reduced to a boundary particle moving in $U(1)$ space. Equivalently, low energy effective action for the complex SYK contains $U(1)$ sigma model. The corresponding action is simply

$$\frac{1}{g_0} \int dt \, (\partial_t \varphi)^2 \, , \varphi \in [0, 2\pi). \tag{4.1}$$

This is just a particle on a circle. The spectrum is $E_n \propto g_0 n^2, n \geq 0$. At low temperatures below the gap $T \ll g_0$ the entropy is zero. For high temperatures it is straightforward to compute the partition function via the Poisson summation formula and the answer for the entropy is

$$S_{extra} = \frac{c}{2} \log \frac{1}{\beta g_0}. \tag{4.2}$$

Extra factor of $c$ comes from having $U(1)^c$. This is an extra piece for the black hole entropy due to its charge fluctuations. Fluctuation entropy behaves as $c \log(\frac{t}{\beta})/2$ and is capped at $t \sim 1/g_0$, hence $S_{\text{gen BH}} \gtrsim S_f$ will be satisfied at high temperatures.

The upshot is that the fluctuation entropy computation is simple as long as $t$ and $l$ are less than $1/g_0$. And moreover $S_f$ might be comparable to $S_{\text{gen BH}}$ only at low temperatures. We first use the above observation to analyze possible paradoxes at zero temperature.

---

[14]In the non-Abelian case the relevant scale in the 't Hooft limit is $g_0 \sqrt{N}$.

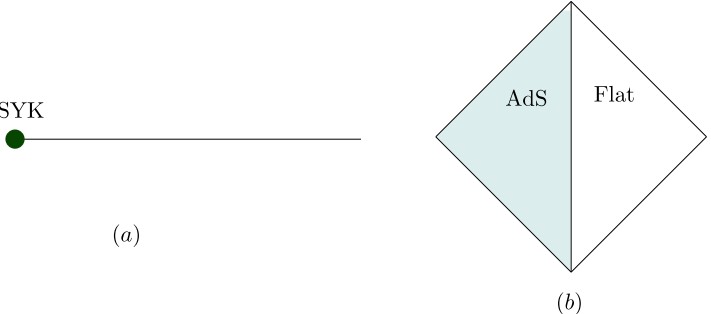

<p style="text-align:center;">(a)</p>
<p style="text-align:center;">(b)</p>

Figure 3: (a) Microscopic setup: SYK dot coupled to a 1+1 CFT in flat spacetime. (b) Corresponding gravity dual. We impose transparent boundary conditions at the $AdS$ boundary such that CFT fields can propagate inside.

## 4.2 A simple paradox at zero temperature

Let us now look at the zero-temperature case, when black hole fluctuation entropy does not contribute. In this case there a simple "bag of gold" type paradox [3, 49]. Here we briefly review it in the absence of any global/gauge symmetries. Imagine coupling 2d CFT to zero-temperature SYK dot. It it dual to JT gravity in Poincare AdS sewed to a flat spacetime region - Figure 3.

SYK has finite entropy but an interval in CFT can be infinitely large, and it cannot be purified by SYK which has entropy $\sim S_0$. A single interval cannot have entropy greater than $S_0$. This expectation is reproduced by islands [3, 49]. We consider interval $[b, +\infty)$ in a CFT and supplement it with an island located at $(-\infty, -a]$ in Poincare AdS. The corresponding generalized entropy is

$$S_{\text{gen}}(a) = S_0 + \frac{\phi_r}{a} + \frac{c}{6} \log \frac{(a+b)^2}{a\epsilon_{UV,b}}. \tag{4.3}$$

For very small $b$, the entropy is extremized at $a^* = \frac{6\phi_r}{c}$ and hence

$$S_{\text{gen}}^* = S_0 + \frac{c}{6} + \frac{c}{6} \log \frac{6\phi_r}{c\epsilon_{UV,b}}. \tag{4.4}$$

Since we took $b$ to be small, this answer should be equal to SYK dot (or black hole) entropy. In other words, we are neglecting the CFT entropy between 0 and $b$. This result indeed agrees with naive expectations of what black hole entropy should be. Extra terms proportional to $c$ should not be too surprising, as propagating fields typically renormalize $S_0$ in JT gravity.

We can be a bit more careful here. We started from a black hole with extremal entropy $S_0$, coupled it to extra matter and found an answer which exceeds $S_0$ by some amount.

Microscopically, we added an interaction between the SYK dot and CFT. We should expect that the resulting entropy computed with gravity should not exceed the (logarithm of) total Hilbert space dimension. It is natural to assume that it cannot be parametrically larger [15] than $S_0$. Basically we are trying to avoid the "bag of gold" paradox by requiring that the extra piece in the entropy does not much exceed $S_0$. It leads to the following parametric bound on $\phi_r$:

$$\phi_r \lesssim c\epsilon_{UV,b} \exp\left(\frac{\mathcal{O}(1)S_0}{c}\right), \tag{4.5}$$

The above argument is the most sharp when we have a dual microscopic model with a finite number of degrees of freedom. However, we will demonstrate that this bound works for $JT$ coming from higher-dimensional gravity theories too,

Now we imagine we have a global or a gauge symmetry. The fluctuation entropy is not sensitive to islands and grows as $(c/2)\log((2/\pi)\log l/\epsilon_{UV,b})$. It can easily exceed $S_0$. In the gauge case it will be capped at $l \sim 1/g_0$, we discuss this in more detail in Sections 4.3, 4.7. Hence we ask that

$$S_0 \gtrsim \frac{c}{2}\log\frac{2}{\pi}\log\frac{1}{g_0\epsilon_{UV,b}}. \tag{4.6}$$

Therefore, we obtain the following bound:

$$g_0 \gtrsim \frac{1}{\epsilon_{UV,b}}\exp\left(-\frac{\pi}{2}\exp\left(\frac{\mathcal{O}(1)S_0}{c}\right)\right). \tag{4.7}$$

So far in the both bounds (4.5), (4.7) $\epsilon_{UV,b}$ is UV cutoff in the non-gravitational region. In principle it can be made arbitrary small. However, since we are computing entanglement entropy for modes which can propagate into the gravitational region, the actual UV cutoff must be larger than the gravitational one, $\epsilon_p$. This is how we obtain the final form of the bounds.

## 4.3  Yang–Mills theory in 2d and twist operators

In our toy model we have a bunch of free fermions(dual to WZW) interacting with YM term. We expect that at the (spacial) scale greater than $1/g_0$, one can neglect the YM kinetic term and the gauge-fields effectively become Lagrange multipliers, "killing" some of the currents (this is known as Goddard–Kent–Olive projection [41,50,51]). We can see this a little bit more explicitly in the $U(1)$ case. The famous Schwinger calculation says that after the bosonization of fermions the gauge field can be integrated out making one the bosons

---

[15]For example, for $q = 4$ SYK with $N$ Majorana fermions, $\log\dim\mathcal{H} = N\log(2)/2 \approx 0.34N$, but $S_0 \approx 0.23N$.

massive with mass $m^2 = \frac{g_0^2}{\pi}$. Let us review this beautiful calculation. We start from a bunch of 2d Dirac fermions $\psi_i, i = 1, \ldots, N$ interacting with a gauge field:

$$\mathcal{L} = \frac{1}{2g_0^2} F^2 + \sum_{i=1}^{N} \overline{\psi}_i \left( \partial_z + A_z \right) \psi_i + \text{(anti-chiral)}. \tag{4.8}$$

After the bosonization we have $N$ real bosons $\phi_i$ which interacts with the gauge field $A$ via the topological current $j_\mu = \frac{1}{2\pi} \sum_i \epsilon_{\mu\nu} \partial_\nu \phi_i$:

$$\mathcal{L} = \frac{1}{2g_0^2} F^2 + \sum_{i=1}^{N} \left( \frac{1}{8\pi} \left( \partial \phi_i \right)^2 + \frac{1}{2\pi} F_{z\bar{z}} \phi_i \right) \tag{4.9}$$

In the above expression we have already integrated by parts to produce $F_{z\bar{z}} \phi_i$. Integration can be switched from $A$ to $F_{z\bar{z}}$ by introducing a delta-function $\delta(F - dA)$. Integrating over $F$ and $A$ yields [16]

$$\mathcal{L} = \frac{1}{8\pi} \sum_{i=1}^{N} \left( \partial \phi_i \right)^2 - \frac{g_0^2}{8\pi^2} \left( \sum_i \phi_i \right)^2 \tag{4.10}$$

It means that the collective bosonic field $\sum_i \phi_i$ is now massive with the mass $m^2 = \frac{g_0^2}{\pi}$. The rest remain massless. In more complicated examples, such as $U(N)$ with fermions in the fundamental, some massless degrees of freedom can remain above the relevant 't Hooft scale $1/(\sqrt{N}g_0)$ but they are not charged [52]. For adjoint fermions all excitations are gapped [53].

Notice that these $\phi_i$ scalars are exactly the same dual scalars we used for computing the fluctuation entropy in Section 2.2 on massless Dirac fermion. Therefore, in the presence of gauge fields the charge distribution and fluctuation entropy can be found by studying the following correlator(with a single or many $\phi$):

$$\left\langle \exp \left( i\alpha \int_R dx \, j_0 \right) \right\rangle = \langle e^{i\frac{\alpha}{2\pi} \phi(l)} e^{-i\frac{\alpha}{2\pi} \phi(0)} \rangle, \tag{4.11}$$

*in a free massive scalar theory.* Obviously, the correlator essentially stops decaying for distances larger than the scalar mass. It means that the fluctuation entropy stops growing. Notice that it happens even at large temperatures. An easy way to see it, is to derive the following expression in the limit $T \gg m$:

$$\langle \phi(l)\phi(0) \rangle \propto T \int dp \sum_n \frac{e^{ipl}}{p^2 + (2\pi Tn)^2 + m^2} \propto T \sum_n \frac{e^{-|l|\sqrt{m^2 + (2\pi Tn)^2}}}{\sqrt{m^2 + (2\pi Tn)^2}} \propto$$

---

[16]If the spacetime is compact we have sectors with different total flux $\int F_{z\bar{z}} = f \in \mathbb{Z}$, here we ignore this subtlety.

$$\propto \frac{T}{m}e^{-|l|m} - 2\log(1 - e^{-2\pi T|l|}), \qquad (4.12)$$

where the first term comes from $n = 0$ mode, and the second term is the result of approximating $\sqrt{m^2 + (2\pi Tn)^2} \approx 2\pi T|n|$ and summing over $n \neq 0$. The actual fluctuation entropy is controlled by $\langle\phi(\epsilon_{\mathrm{uv}})\phi(0)\rangle - \langle\phi(l)\phi(0)\rangle$. This quantity indeed saturates at $T/m$ for large intervals.

## 4.4  Entanglement entropy and gauge symmetry

In principle, we can consider a model where gauge fields exist in both gravitational and non-gravitational regions. However, as has been emphasized in a lot of papers [54–58], entanglement entropy is subtle in gauge theories. The problem is that if we have complementary regions $R, \overline{R}$ the total Hilbert space does not factorize: $\mathcal{H} \neq \mathcal{H}_R \otimes \mathcal{H}_{\overline{R}}$. The most direct way to see this is lattice gauge theory: in this case gauge degrees of freedom are strings. A string crossing a boundary of $\partial R = \partial \overline{R}$ does not belong to either $\mathcal{H}_R$ or $\mathcal{H}_{\overline{R}}$. This results in an ambiguous relation between subregions and algebras, but does not prevent us from uniquely[17] defining entanglement entropy [58]. In the Abelian case the full Hilbert space has the following decomposition:

$$\mathcal{H} = \bigoplus_q \mathcal{H}_{R,q} \otimes \mathcal{H}_{\overline{R},q}, \qquad (4.13)$$

where $q$ is the flux through the boundary of $R$(the total charge inside). To put it differently, for a fixed $q$ the algebra of observables does not have a center. Now in each sector $q$ we can compute the entanglement entropy $S_R(q)$. These are exactly the symmetry resolved entropies introduced in Section 2. We sum over them and add the contribution $S_f$ from the charge distribution $p_R(q)$. We arrive at the eq. (2.3):

$$S_R = \sum_q p_R(q)S_R(q) - \sum_q p_R(q)\log p_R(q), \qquad (4.14)$$

however in the gauge case this is the definition of the entropy, rather than a decomposition of an independently defined quantity.

The necessity to project on a given charge can be seen in the path integral too. Consider a standard path-integral for computing $n-$Renyi of $R$. Figure 4 illustrates the $n = 2$ case. The spacetime has $n$ branches connected at $R$. It has a non-trivial topology: there are non-contractible cycles. So according to the rules of gauge theory we have to integrate over

---

[17]at least in the abelian case. For some extra subtleties regarding the non-Abelian case we refer to [57]. Our computations in the non-Abelian case follow this reference and include the term $\log\dim_\lambda$.

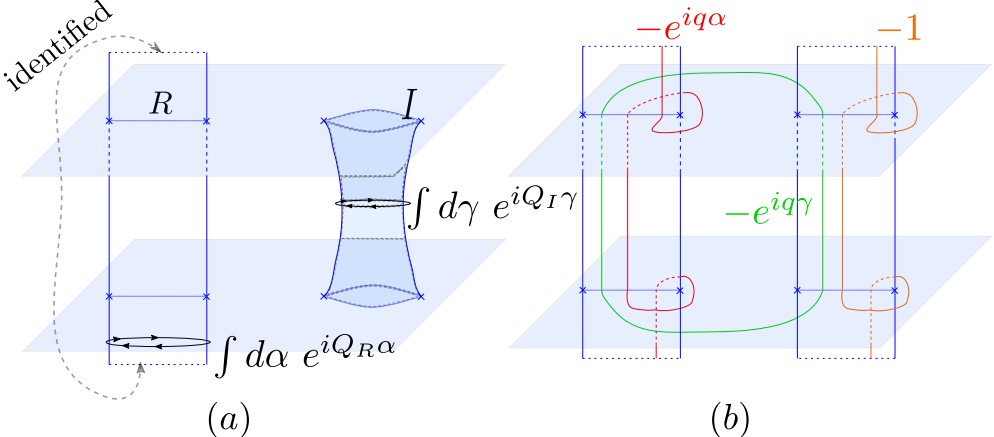

Figure 4: (a) Replicated $n = 2$ geometry for the interval $R$ and the island $I$. A theory on such geometry may have fluxes in $R$(denoted by $\alpha$) and around the replica wormhole(denoted by $\gamma$). (b) A more conventional CFT picture with cuts on a Riemann surface. Different colors represent different closed cycles. Minus signs are related to Fermi statistics. Charge $q$ matter fields going around the red cycle acquire a phase $q\alpha$. It can be used to impose zero charge constraint for the corresponding interval. Notice that the flux $e^{iQ_I\gamma}$ is now inserted into both replica copies, because particles going back-and-forth within the wormhole should be insensitive to $\gamma$(orange path). However, fields going around the green path acquire the phase $q\gamma$. Integration over $\gamma$ prohibits the charged particles to go between $R$ and $I$, enforcing the charge conservation. Notice that there is no analogue of $\alpha$ Aharonov–Bohm flux for the island.

all possible holonomies of the matter fields(insertion of Wilson lines). For a single interval, there are only red $\alpha$ cycles on Figure 4. It corresponds exactly to the Aharnov–Bohm flux $\alpha$ introduced in Section 2.1. Integration over $\alpha$ sets the charge of the corresponding interval to zero. Contributions from other sectors can be obtained by inserting $e^{-iq\alpha}$.

$$\int d\alpha \; e^{-iq\alpha} Z_n(\alpha) = \mathcal{Z}_n(q_R = q). \tag{4.15}$$

The upshot is that the non-factorisation of the Hilbert space can be seen on the level of the replica path integral from the presence of extra holonomies.

### 4.5   Islands: extra holonomies

Now we want to understand how Wilson lines enter into the island prescription:

$$S(\rho_R^{exact}) = \min \; \text{ext}_I \left( \frac{A_I}{4G_N} + S(\rho_{R\cup I}^{semi}) \right). \tag{4.16}$$

Presumably it should imply the equality of modular Hamiltonians: $K_R^{exact} = K_{R\cup I}^{semi}$. This fact was explored in [59] to pull the information from the island. Another consequence of this fact is the violation of global symmetries [23]. The goal of this Section is to understand what is $\rho_{R\cup I}^{semi}$ in the gauged case and how Wilson lines save the symmetry.

For replica geometry with two intervals we have extra $\gamma$ holonomies, Figure 4. What do they correspond to? Suppose we have two intervals, $R$ and $I$. Total charge $Q_R+Q_I$ commutes with the density matrix $\rho = \rho(R\cup I)$. However, performing a symmetry transformation only on $R$ does not leave it invariant:

$$e^{iQ_R\alpha}\rho^{semi}(R\cup I)e^{-iQ_R\alpha} \neq \rho^{semi}(R\cup I) \tag{4.17}$$

And vice versa for $R \leftrightarrow I$. This is one way to see the violation of global symmetries because of the islands. One universal way to make something invariant under a group action is to average over all possible actions. We can make it gauge invariant again by the following proposition:

$$\rho_{gauge}^{semi}(R\cup I) := \int_{-\pi}^{\pi} d\gamma \; e^{iQ_I\gamma}\rho^{semi}(R\cup I)e^{-iQ_I\gamma}. \tag{4.18}$$

In the rest of this Section we will argue that this definition naturally arises from integrating over the Wilson loop going around the wormhole throat.

Before we demonstrate various good properties of $\rho_{gauge}^{semi}$, let us say right away that definition (4.18) looks like a coarse-graining procedure[18], as it can map pure states into mixed states. This is a reasonable concern. However, we do not find any explicit problems, the resulting entanglement entropy expressions respect the purity of the state, as we discuss at the end of this Subsection. The path integral naturally produces this prescription, and it is not clear what other prescriptions can save us from charge non-conservation [19].

The good properties of $\rho_{gauge}^{semi}(R)$ are the following:

- Normalization: $\text{Tr}\left(\rho_{gauge}^{semi}(R)\right) = 1$.

- It has a positive spectrum.

---

[18]Similar density matrix was studied in [23] within the "west coast" model and it was found that the connected saddle never dominates.

[19]In principle, one can concoct other prescriptions which enforce the charge conservation. For example, setting island charge to zero: $\rho_{q_I=0}(R \cup I) = \int d\gamma_1 d\gamma_2 e^{i\gamma_1 Q_I}\rho(R\cup I)e^{i\gamma_2 Q_I}$. However, this does not follow from replica wormholes. Moreover, this density matrix does not satisfy eq. (4.20), making the evaluation of local observables ambiguous. Curiously, $\rho_{q_I=0}(R\cup I)$ does not have a problem with growing fluctuation entropy.

- It preserves the charge in the region $R$:

$$[\rho_{gauge}^{semi}(R \cup I), Q_R] = 0 \tag{4.19}$$

- Tracing out $I$ yields again $\rho_R$ in CFT vacuum:

$$\mathrm{Tr}_I \, \rho_{gauge}^{semi}(R \cup I) = \rho^{semi}(R). \tag{4.20}$$

Let us stress that $\rho^{semi}(R \cup I)$ is evaluated using standard CFT rules if the coupling is small(which we are assuming in this Section). This equation holds even if the gauge field is present in the gravitational region only.

We want to understand what replica geometry we expect from $\int d\gamma \ e^{i\gamma Q_I} \rho^{semi} e^{-iQ_I\gamma}$. Taking it to $n-$power, we get:

$$\int_{-\pi}^{\pi} d\gamma_1 \ldots d\gamma_n \, \mathrm{Tr} \left( \rho^{semi} e^{i(\gamma_2 - \gamma_1)Q_I} \rho^{semi} e^{i(\gamma_3 - \gamma_2)Q_I} \ldots \rho^{semi} e^{i(\gamma_1 - \gamma_n)Q_I} \right) \tag{4.21}$$

So the fields(with unit charge) going around interval $I$ cut are transformed under the following matrix:

$$T_I = \begin{pmatrix} 0 & e^{i(\gamma_2 - \gamma_1)} & 0 \ldots & \\ 0 & 0 & e^{i(\gamma_3 - \gamma_2)} & 0 \ldots \\ \ldots & & & \\ (-1)^{n+1} e^{i(\gamma_1 - \gamma_n)} & 0 & & \ldots \end{pmatrix} \tag{4.22}$$

It is not difficult to see that this monodromy matrix correctly reproduces the matter boundary conditions illustrated by Figure 4 (b), green and orange cycles. These holonomies stop the charge from jumping between the intervals. Geometrically, $\gamma$ corresponds to the time component of the gauge connection around the wormhole throat, Figure 4 (a).

Notice that this matrix does not correspond to the monodromy matrix from inserting Aharonov–Bohm fluxes. Since that one would have equal phases for going between the sheets:

$$T_{I,flux} = \begin{pmatrix} 0 & e^{i\gamma/n} & 0 \ldots & \\ 0 & 0 & e^{i\gamma/n} & 0 \ldots \\ \ldots & & & \\ (-1)^{n+1} e^{i\gamma/n} & 0 & & \ldots \end{pmatrix} \tag{4.23}$$

In some sense the two are complimentary: Aharonov–Bohm flux (4.23) fixes "the center of mass", whereas $\gamma-$holonomies (4.22) correspond to the motion around it.

Finally, let us comment on state purity. In the setup of Figure 1, the entanglement

entropy of $R$ must be equal to the entropy of $[P_1, P_2]$ interval. Computation of $S([P_1, P_2])$ will contain the $[P_3, P_4]$ island as well, leading to the same two interval $[P_1, P_3] \cup [P_4, P_2]$ answer. One confusing point however is that $[P_1, P_3] \cup [P_4, P_2]$ does not contain $I$, so how are we supposed to act with $e^{iQ_I\gamma}$ on $\rho^{semi}([P_1, P_3] \cup [P_4, P_2])$? The resolution[20] is that in gauge theories we can use Gauss law to rewrite $Q_I$ as difference of electric fields at points $P_3, P_4$. After that we can reshuffle the replicas and obtain the entropy of $R$.

## 4.6 Estimating the entropy

Having proposed the definition of $\rho_{gauge}^{semi}(R \cup I)$, we want to compute its entropy in some simple cases. Unfortunately, it is very hard: replicated fields going around $I$ branch cut undergo the transformation (4.22), whereas region $R$ has the same monodromy matrix, but with all $\gamma$ equal to zero: $\gamma_1 = \cdots = \gamma_n = 0$. These two matrices do not commute.

Using the concavity of von Neumann entropy one can show that

$$S(\rho_{gauge}^{semi}(R \cup I)) \geq S(\rho^{semi}(R \cup I)), \tag{4.24}$$

so the entropy will increase. On physical grounds this should be expected for any density matrix $\rho^{semi}(R \cup I)$ preserving the charge in $R$: preserving the charge in $R$ is equivalent to prohibiting the charge flow through the wormhole. The job of the island is to purify region $R$ and this charge constraint makes this job more difficult.

In Appendix C we argue that

$$\max\left(S_f(R), S_f(I)\right) \leq S(\rho_{gauge}^{semi}(R \cup I)) \leq S(\rho^{semi}(R \cup I)) + \min\left(S_f(R), S_f(I)\right). \tag{4.25}$$

The argument there uses only the definition (4.18) of $\rho_{gauge}^{semi}$ and does not invoke any assumptions about gravity or gauge fields.

In the kinematics we are interested in, $S(\rho^{semi}(R \cup I))$ does not grow with the interval length(this is the main consequence of the island prescription). Hence we essentially have a tight bound on $S(\rho_{gauge}^{semi}(R \cup I))$. Therefore we conjecture that in the OPE limit(long island and interval $R$) it actually behaves as

$$S(\rho_{gauge}^{semi}(R \cup I)) \approx S(\rho^{semi}(R \cup I)) + S_f(I). \tag{4.26}$$

One heuristic argument for this can be the following: if $I, R$ are long we can study the OPE limit, when the twist operators of $I, R$ fuse together. This OPE produces identity(and

---

[20]We are grateful to Geoff Penington for pointing this out.

descendants). Insertion of $e^{i\gamma Q_I}$ is a line operator. Descendants of identity near the ends of $I$ can alter its expectation value relative to the vacuum, but for long $I$ it should be a small correction. Therefore we are left with the insertions of $e^{i\gamma Q_I}$ which produce $S_f(I)$, by eq. (C.7).

We conclude this Section by asking: can the island adjust to the extra $S_f$ term in eq. (4.26)? Naively, it seems that by making $I$ shorter than $R$, we can try to cap the growth of the entropy. We should say right away that eq. (4.26) was conjectured in the OPE limit, when $S_f(R) \approx S_f(I)$. Hence by taking eq. (4.26) literally and allowing the length of the island to be slightly different from $R$, we really exceed the accuracy of this equation. So we see the calculation below as a qualitative probe of how the island configuration might change once we include the holonomies.

Our conclusion is that a small change in the island length can not stop $S_f$ from growing. This happens because the standard CFT answer grows much faster and a tiny variation in its length can make the generalized entropy extremal again. Let us illustrate this statement by a simple setup in the gravitationally prepared state of Section 3.2. We assume there is no bra-ket wormhole, but the calculation below can be easily generalized to that case. The radiation region $R$ is the interval $[0, l]$, whereas the island $I$ is $[-iz + d/2, -iz + l - d/2]$. The setup is illustrated by Figure 2(a). Note that we allowed the island to have a different length now. The relevant terms in the generalized entropy are

$$\frac{c}{6} \log \left( z^2 + \frac{d^2}{4} \right) + \frac{c}{2} \log \log(l - d). \tag{4.27}$$

This should hold as long as $d \ll l$. Extremizing over $d$ yields

$$\frac{d}{d^2/4 + z^2} = \frac{6}{(l - d) \log(l - d)}. \tag{4.28}$$

As a function of $d$, this equation has two solutions, but the actual minimum of the entropy is when $d$ is small, namely

$$d^* \approx \frac{6z^2}{l \log l} \ll l, \tag{4.29}$$

since $z$ is of order 1 for long intervals. As we have promised, the change in the island length is very small. We checked that the same conclusion holds for the eternal black hole.

## 4.7 Fluctuation entropy for large gauge coupling

In section 4.6, we estimated the entropy when the gauge coupling $g_0 \to 0$. Here we consider the opposite case where the gauge coupling is large compared to the regularized dilaton. The

goal is to show that the fluctuation entropy does not grow indefinitely. We assume that the gauge field only exists in the gravitational region. The matter content in the bath region is $c$ free fermions. In the gravity region, the matter Lagrangian is given by $c$ gauge fields (with YM terms) which are coupled to each fermion individually. The fermions satisfy transparent boundary conditions, while we impose Dirichlet boundary conditions for the gauge fields at the interface. This is similar to the boundary conditions imposed for the metric.

Before going into the calculation, let us present the main physical picture. As mentioned in Section 4.3, the massless Schwinger model is confined and the content of theory is neutral mesons. The value of mass for each fermion is determined by the gauge coupling $m^i = g_0^i/\sqrt{\pi}, i = 1, \cdots, c$ [21]. If the Compton wavelength associated to this mass is parametrically smaller than the length of the interval $l$, then we find that the fluctuation entropy does not grow anymore. This is obvious if the gauge symmetry is included everywhere, because the twist fields $e^\phi$ become massive. However, in our case we included gauge fields $A_\mu^i$ in the gravitational region only. Outside the fermions are still massless. So it is not obvious that the fluctuation entropy will stop growing and the paradox gets resolved. Below we are going to demonstrate that including the gauge fields in the gravitational region only is indeed enough.

The key observation is that for large coupling (interval length is much bigger than $1/g_0^i, i = 1, \cdots c$), the gauge fields effectively change the matter boundary conditions from the transparent to the Dirichlet. This means we have to calculate the fluctuation entropy for fermions with Dirichlet boundary condition. This is a well-known case for the entanglement entropy [60,61] and has been analyzed recently for the $U(1)$ symmetry-resolved entropy in [12]. Here we only work in the limit that the Compton wavelength is small and we leave the analysis of an intermediate gauge coupling regime to the future.

### 4.7.1   Eternal black hole

In order to have a well-defined variational principle, we impose the Dirichlet boundary condition for the gauge fields along the boundary

$$A_\tau^i\big|_{\sigma=\epsilon} = 0. \tag{4.30}$$

This boundary condition is consistent with having a gauge symmetry inside the disk and a global symmetry outside [62]. Note that this boundary condition makes the two-dimensional theory of pure gauge field trivial. However, here we look for the simplest theory of gauge

---

[21]In order to have the $U(1)^c$ gauge theory, we consider slightly different couplings for each gauge field. However, their exact values are not important for this section.

fields coupled to matter to avoid the issue found in section 3.1.

In the large coupling $g_0^i$ limit, the Maxwell term in (4.8) is negligible and by integrating over in the path integral $A_\mu^i$, we find

$$j_\mu^i = \frac{1}{2\pi}\epsilon_{\mu\nu}\partial_\nu\phi_i = 0, \qquad i = 1, \cdots c \qquad \text{inside the unit disk.} \tag{4.31}$$

This means each fermion satisfies the Dirichlet boundary conditions at the interface. In order to compute $Z_1(\alpha)$ for each fermion, we consider the correlation functions of vertex operators $\mathcal{V}_\alpha^i(x) = \exp\left(\frac{i}{2\pi}\alpha\phi^i(x)\right)$. The coordinates here are the same as coordinates in Section 3.1. With a map to a plane with $z = \frac{ie^{2\pi y/\beta}+1}{e^{2\pi y/\beta}+i}$, one finds

$$Z_1(\alpha) = c_\alpha\langle\mathcal{V}_{-\alpha}(P_1)\mathcal{V}_\alpha(P_2)\rangle_D = c_\alpha|z'(y_1)z'(y_2)|^{\Delta_\alpha}\left(\frac{|z_1-\bar{z}_2||z_2-\bar{z}_1|}{|z_1-z_2||\bar{z}_1-\bar{z}_2||z_1-\bar{z}_1||z_2-\bar{z}_2|}\right)^{\Delta_\alpha}$$

$$= c_\alpha\left[\frac{\pi^2}{\beta^2\sinh^2(2\pi b/\beta)}\frac{\cosh\left(\frac{2\pi}{\beta}(t+b)\right)\cosh\left(\frac{2\pi}{\beta}(t-b)\right)}{\cosh^2\left(\frac{2\pi}{\beta}t\right)}\right]^{\Delta_\alpha}, \tag{4.32}$$

$\Delta_\alpha = \left(\frac{\alpha}{2\pi}\right)^2$. The label $D$ denotes the Dirichlet boundary condition. This implies that the effective length of the interval is given by

$$Z_1(\alpha) = c_\alpha l_{\text{eff}}^{-2\Delta_\alpha},$$

$$l_{\text{eff}}^2 = \frac{\beta^2}{\pi^2}\sinh^2(2\pi b/\beta)\frac{\cosh^2(2\pi t/\beta)}{\cosh\left(\frac{2\pi}{\beta}(t+b)\right)\cosh\left(\frac{2\pi}{\beta}(t-b)\right)} \underset{t\gg b}{\simeq} \frac{\beta^2}{\pi^2}\sinh^2(2\pi b/\beta). \tag{4.33}$$

Unlike the case for the transparent boundary condition, $Z_1(\alpha)$ does not decrease indefinitely as a function of time. As a result, the fluctuation entropy remains finite for all times. If we assume $b \gg \beta$ in order to approximate $\mathcal{Z}_1(q)$ by a Gaussian integral over an infinite line, the fluctuation entropy for the $U(1)^c$ gauge theory is given simply by

$$S_f = -\sum_q p(q)\log p(q) = \frac{c}{2}\log\left(\frac{2}{\pi}\log\left(\frac{\beta}{\pi}\sinh(2\pi b/\beta)\right)\right). \tag{4.34}$$

### 4.7.2 Bra-ket wormhole

This case is very similar to the eternal black hole example. The coordinates are the same as section 3.2. This time, we impose the following boundary conditions for the gauge fields

$$A_x^i(z=0,x) = A_x^i(z=2\tau,x) = 0, \qquad \lim_{x\to\pm\infty}A_z^i(z,x)\to 0, \qquad i = 1,\cdots c. \tag{4.35}$$

These boundary conditions render the variation of action well-defined.

Working in the large coupling limit, we find the same condition as (4.31). Therefore, $Z_1(\alpha)$ is effectively computed by imposing the Dirichlet boundary conditions for the $c$ fermion modes along $z = 0, 2\tau$ surfaces. Since these modes remain massless in the outside region, we again compute the two-point function of vertex operators by mapping the problem to the upper half plane and use the method of images. The answer for the charged moment of each fermion for points $P_1 = (z = \tau, x = -l/2), P_2 = (z = \tau, x = l/2)$ is

$$Z_1(\alpha)/Z_1(0) = c_\alpha \langle \mathcal{V}_{-\alpha}(P_1)\mathcal{V}_\alpha(P_2)\rangle_D = c_\alpha \left( \left(\frac{\pi}{2\tau}\right) \coth \frac{\pi l}{4\tau} \right)^{\frac{\alpha^2}{2\pi^2}}$$

$$\underset{l \gg \tau}{\simeq} c_\alpha \left(\frac{\pi}{4\tau}\right)^{\frac{\alpha^2}{2\pi^2}} \tag{4.36}$$

Therefore, the probabilities $p(q)$ and the fluctuation entropy $S_f$ remain finite for large intervals.

# 5 Further comments

## 5.1 Perturbative and non-perturbative corrections

We should also discuss the stability of our results against perturbative and non-perturbative corrections.

The perturbative ones can arise, for example, if the symmetry is gauged and we add a YM term, or from the space-time fluctuations. As for the former, we have discussed in Section 4, that in 2d eq. (2.30) *should be valid for any $\alpha$ as long as $l$ is smaller than the inverse coupling.* It implies that the distribution $p(q)$ has the following form:

$$p(q) = p_0(q)(1 + \varepsilon(q)), \tag{5.1}$$

where $p_0(q)$ is the Gaussian answer, eq. (2.21), and $\varepsilon(q) \ll 1$ is a small correction. For simplicity we will explicitly consider the Abelian case, but similar considerations can be applied for the non-Abelian case as well. Correction $\varepsilon(q)$ is small for any $q$. Then the first-order correction to the Shannon entropy $S_f$ is

$$\delta S_f = -\sum_q p_0(q)\varepsilon(q) \log p_0(q) - \sum_q \varepsilon(q)p_0(q). \tag{5.2}$$

Since $\varepsilon(q)$ is small, $\delta S_f \ll S_f$. Here it is actually important that $p_0(q)$ is non-zero for all $q$.

Non-perturbative corrections are much more interesting. They can arise from including extra gravitational saddles in the gravity region. Fluctuation entropy can be extracted from a certain two-point function, eq. (2.30), and the reader may wonder about the similarities and differences between growing fluctuation entropy and the Maldacena's eternal black hole paradox [63] in AdS. In thermal AdS background, without any coupling to an external bath, the two point functions decay exponentially forever which is inconsistent with unitarity when the correlator becomes smaller than $e^{-\mathcal{O}(1)S_0}$ [63]. In our case, the indefinite growth of the fluctuation entropy may also seem to follow from the unbounded decay of vertex operators on the black hole coupled to the bath background. The difference from the previous argument is that the correction $\varepsilon(\alpha)$ to $\langle e^{iQ_R\alpha}\rangle$:

$$\langle e^{iQ_R\alpha}\rangle = \langle e^{iQ_R\alpha}\rangle_0 + \varepsilon(\alpha) \tag{5.3}$$

is small, but can actually be big compared to free $\langle e^{iQ_R\alpha}\rangle_0$ answer.

We would like to demonstrate that small non-perturbative corrections to eq. (2.30) which prevent its decay for very long $l$ cannot lower the fluctuation entropy much. We provide two arguments for this. A simpler, but a less general one, assumes that the non-perturbative corrections come from other space-time saddles, as baby universes for example. Recently this setup was explored in [34] and it was demonstrated within JT gravity that Euclidean saddles with extra handles("lid" geometries) can indeed stop correlators from unbounded decay. In this case the answer for $\langle e^{iQ_R\alpha}\rangle$ looks as follows:

$$\langle e^{iQ_R\alpha}\rangle = \frac{1}{1+e^{-2S_0}}\left(\langle e^{iQ_R\alpha}\rangle_0 + e^{-2S_0}\langle e^{iQ_R\alpha}\rangle_{\text{lid}}\right), \tag{5.4}$$

where $2S_0$ is the gravity action and the denominator comes from properly normalizing the partition function. The extra "lid" contribution might not decay for large $l$, hence comparing this expression with eq. (2.30) we see that it might become dominant for $k\alpha^2\log(l) \gtrsim S_0$. *However, the crucial point is that "lid"(or other geometries) can still be used as semi-classical backgrounds for quantizing our matter QFT.* Hence, after the Fourier transform in $\alpha$, this extra contribution will produce a positive probability distribution $p_{\text{lid}}(q)$ for charges:

$$p(q) = \frac{1}{1+e^{-2S_0}}\left(p_0(q) + e^{-2S_0}p_{\text{lid}}(q)\right). \tag{5.5}$$

What remains to do is to use the concavity of Shannon entropy:

$$S(\lambda p_0(q) + (1-\lambda)p_{\text{lid}}(q)) \geq \lambda S(p_0(q)) + (1-\lambda)S(p(q)_{\text{lid}}) \geq \lambda S(p_0(q)) = \frac{1}{1+e^{-2S_0}}S(p_0(q)), \tag{5.6}$$

where $\lambda = 1/(1 + e^{-2S_0})$. Hence the fluctuation entropy cannot decrease more than by a factor of $1/(1 + e^{-2S_0})$.

A more universal but tedious argument is to directly analyse how small corrections at large $l$ in eq. (2.30) propagate to Shannon entropy. Now the crucial point is that even if the length $l$ of the interval is huge, the result (2.30) for the two point functions is still valid as long as $\alpha$ is not too large. We would like to demonstrate that the entropy is indeed dominated by such "small" $\alpha$ values. To be concrete, consider $c$ Dirac fermions coupled to $U(1)^c$. Then we have $c$ different $\alpha$ angles and the correlator (2.30) is

$$\langle e^{i \sum_{a=1}^{c} \alpha_a Q_{R,a}} \rangle_0 = \exp\left( -\frac{2 \log(l/\epsilon_{\mathrm{uv}})}{(2\pi)^2} \left( \sum_{a=1}^{c} \alpha_a^2 \right) \right). \tag{5.7}$$

We assume that the correction has the form[22]:

$$\langle e^{i \sum_{a=1}^{c} \alpha_a Q_{R,a}} \rangle = \langle e^{i \sum_{a=1}^{c} \alpha_a Q_{R,a}} \rangle_0 + e^{-2S_0} f(\alpha), \tag{5.8}$$

where $f(0) = 0$(normalization) and $|f(\alpha)| \leq 1$. We do not need to assume anything more about $f(\alpha)$. Performing a Fourier transform we obtain that

$$p(q) = p_0(q) + e^{-2S_0} f(q), \tag{5.9}$$

where $|f(q)| \leq 1$, but can have any sign. Distribution $p_0(q)$ is the generalization of eq. (2.21):

$$p_0(q) = \prod_{a=1}^{c} \sqrt{\frac{\pi}{2 \log l/\epsilon_{\mathrm{uv}}}} e^{-\frac{\pi^2 q_a^2}{2 \log(l/\epsilon_{\mathrm{uv}})}}. \tag{5.10}$$

$e^{-2S_0} f(q)$ is a small correction as long as $q_a^2/\log(l/\epsilon_{\mathrm{uv}}) \lesssim S_0/c$ for all $a$. However, it is easy to see that the fluctuation Shannon entropy is dominated by $q_a^2/\log(l/\epsilon_{\mathrm{uv}}) \lesssim 1$. Hence as long as $S_0/c$ is big, corrections to it are suppressed by $c/S_0$.

In the models studied in the literature(and this paper is not an exception), the ratio $S_0/c$ is fixed and is assumed to be big. So the correction is indeed small. However, we would like to mention that gauge theories often acquire fractional non-perturbative effects. Examples include infrared renormalons in 4d YM [64] and 2d $\mathbb{C}P^N$ model [65]. Repeating the same analysis for "fractional" gravity contributions of order $e^{-S_0/c}$ trivially yields that now $p(q)$ is affected for $q_a^2/\log(l/\epsilon_{\mathrm{uv}}) \gtrsim S_0/c^2$ and since $S_0/c^2$ is small, the correction to Shannon entropy might be big. However, we are not aware of any indications why order $e^{-S_0/c}$ corrections exist.

---

[22]This is the same form as eq. (5.4), as the factor $e^{-2S_0} \langle e^{i\alpha Q_R} \rangle_0 / (e^{-2S_0} + 1)$ can be moved into $e^{-2S_0} f(\alpha)$.

## 5.2 Ensemble average resolution?

We can rephrase the above results in terms of coarse-graining. By coarse-graining we mean that we have some exact $\rho_{fine}$ and approximate $\rho_{coarse}$, which reproduces the expectation values of simple operators $O$ up to non-perturbative corrections:

$$\text{Tr}\left(O\rho_{fine}\right) = \text{Tr}\left(O\rho_{coarse}\right) + \mathcal{O}\left(e^{-S_0}\right). \tag{5.11}$$

Computations using a disk saddle should be viewed as computations in $\rho_{coarse}$. von Neumann entropy $S\left(\rho_{coarse}\right)$ of $\rho_{coarse}$ can be big. In fact, we should maximise it over all possible $\rho_{coarse}$ satisfying eq. (5.11). However, $S\left(\rho_{fine}\right)$ should not exceed Bekenstein–Hawking entropy. One of the main claims of this paper is that eq. (5.11) is not that innocent: for operator $O = e^{iQ_R\alpha}$ it puts a lower bound on both $S\left(\rho_{coarse}\right)$ and $S\left(\rho_{fine}\right)$. The previous subsection highlights that this lower bound is not sensitive to $\mathcal{O}\left(e^{-S_0}\right)$ corrections. To say it another way, "fine" vs "coarse" does not save us from very big fluctuation entropy.

Recently, it was proposed in the literature that semiclassical gravity is really dual to the ensemble average of quantum field theories. Does it help us with big fluctuation entropy? Surprisingly, it can. Let us denote the ensemble averaging as $[\cdot]_J$. Now all the observables should be treated as random variables with respect to some probability measure over the ensemble. For example, 2d JT gravity we studied can be thought of as dual to disordered Sachdev–Ye–Kitaev(SYK) model. Then all we compute in gravity include the ensemble averaging $[\cdot]_J$. *Crucially, even the Fourier transform of $\langle e^{iQ_R\alpha}\rangle$ yields the averaged charge distribution:*

$$\int d\alpha \; e^{-i\alpha q}\langle e^{iQ_R\alpha}\rangle_{gravity} = [p(q|J)]_J \equiv p(q), \tag{5.12}$$

where $p(q|J)$ is the conditional $q$-distribution, with the ensemble parameter $J$ fixed. Once averaged over $J$ it yields $p(q)$ which we have computed in this paper. This averaged distribution is good in sense that it will reproduce *exactly* all the simple expectation values $g(Q)$ we can compute in gravity due to linearity:

$$\langle g(Q)\rangle_{gravity} = [\sum_q g(q)p(q|J)]_J = \sum_q g(q)[p(q|J)]_J. \tag{5.13}$$

Here we used the fact that the charge operator $q$ has a non-perturbative definition independent of $J$.

However, the Shannon entropy of $[p(q|J)]_J = p(q)$ can be very far away from the averaged

Shannon entropy. Due to concavity of the Shannon entropy, the later one is smaller:

$$S_f\left(p(q)\right) \geq [S_f(p(q|J))]_J. \tag{5.14}$$

Presumably, Bekenstein–Hawking entropy should be compared to the averaged fluctuation entropy $[S_f(p(q|J))]_J$. In classical statistics it is known as conditional entropy. We expect this quantity to be bounded and does not grow indefinitely. However, it would mean that the *entropy of ensemble* is unbounded. It follows from the following simple equality for conditional entropies:

$$[S_f(p(q|J))]_J = \langle S_{\text{ens}}(p(J|q)) \rangle_q + S_f(p(q)) - S_{\text{ens}}(p(J)). \tag{5.15}$$

Let us explain $S_{\text{ens}}$ in this equality. We assume that we have some ensemble distribution $p(J)$ and conditional distribution $p(q|J)$. Standard Shannon entropy of $p(J)$ is denoted as $S_{\text{ens}}(p(J))$. Then by the definition of conditional probability, they define a joint probability distribution $p(q, J)$:

$$p(q|J)p(J) = p(q, J). \tag{5.16}$$

From it we can extract the conditional distribution $p(J|q)$ and the corresponding conditional entropy $\langle S_{\text{ens}}(p(J|q)) \rangle_q$:

$$\langle S_{\text{ens}}(p(J|q)) \rangle_q = -\int dq \; p(q) \left( \int dJ \; p(J|q) \log p(J|q) \right). \tag{5.17}$$

For discrete distributions $\langle S_{\text{ens}}(p(J|q)) \rangle_q$ is trivially positive [23]. If we assume that this quantity is also positive for continuous distributions, we get the following inequality:

$$S_{\text{ens}}(p(J)) \geq S_f(p(q)) - [S_f(p(q|J))]_J. \tag{5.18}$$

Assuming $[S_f(p(q|J))]_J$ does not grow with the interval length, we see that $S_{\text{ens}}(p(J))$ must be unbounded.

Of course, each term in the original equality (5.15) should have local counterterms added to make them UV-finite. However, these counterterms are local so they would not be able to cap $S_f(p(q))$ which grows with volume. So similar to our discussion on the fluctuation

---

[23]Without an extra physical input, there is a toy model when $\langle S_{\text{ens}}(p(J|q)) \rangle_q$ can be made very negative. Consider $p(q, J) \propto \exp\left(-AJ^2 + 2BJq - Cq^2\right)$. If the parameter $B$, which represents the correlation between $q$ and $J$, has a strong dependence on the interval length $l$, one can make $p(J)$ $l$-independent and $S_{\text{ens}}(p(J))$ finite, $[S_f(p(q|J))]_J$ finite, $S_f(p(q))$ positive and growing, but $\langle S_{\text{ens}}(p(J|q)) \rangle_q$ *arbitrary negative* for large intervals by tuning $A, C$.

entropy and von Neumann entropy, inequality (5.18) should be understood parametrically: $S_f(p(q))$ contains a growing $\log(l/\beta)$ term, so $S_{\mathrm{ens}}(p(J))$ must also grow with the volume.

Another small issue is related to the definition of $S_{\mathrm{ens}}(p(J))$ and $S_{\mathrm{ens}}(p(J|q))$ for continuous distributions. One cannot directly compute them by introducing a discrete cutoff $\Delta J$, as the resulting expression will contain $\log(\Delta J)$ which diverges for $\Delta J \to 0$. However, the key eq. (5.15) contains the difference $S_{\mathrm{ens}}(p(J)) - \langle S_{\mathrm{ens}}(p(J|q))\rangle_q$ where $\log(\Delta J)$ cancels out. So instead of $\Delta J$ one can use an arbitrary "renormalization" scale $\mu$ in order to have the right dimension inside the $\log(p(J)\mu)$.

## 5.3  The bound in concrete examples

The setup we studied, namely 2d JT gravity with matter and gauge symmetry, naturally emerges in two cases: near-horizon limit of 4d magnetic black holes and complex SYK model. It is very interesting to see if our proposed bound makes sense in these models. In this Section we will see that in both cases the bound is satisfied.

### 5.3.1  Complex SYK model

A single complex SYK model has two soft modes: time-reparametrizations which are governed by Schwarzian action and represent the gravity sector, and $U(1)$ sigma-model action [66]. The later can be thought of as 2d Maxwell term [67] and we identify the sigma-model coupling $K$ with 2d YM coupling $g_0$ as [24]

$$K \propto \frac{1}{g_0}. \tag{5.19}$$

A single complex SYK gives rise only to one $U(1)$ symmetry. In order to get a large $U(1)^c$ group we simply take $c$ coupled complex SYK models. In practice, it has to be done with care as coupled SYK models tend to have spontaneous symmetry breaking [68, 69] and are not always described by Schwarzian action [70, 71]. We assume that all $c$ SYK systems are coupled in a way that this does not happen. One way to do it is to avoid the emergence of the mixed correlators between the different SYK dots. This can be achieved by making all the disorder couplings independent. Making them equal in magnitude excludes a known case when Schwarzian is non-dominant [70, 71]. Hence we assume that all four-fermion couplings (within each SYK and between different SYK) are of the same order $J$. This seems rather

---

[24] A simple way to relate the sigma model action $K \int d\tau (\partial\varphi)^2$ to the bulk YM is to compare the gap between the charged sectors. From the Schwinger effect in Section 4.3 we expect the gap in YM to be $g_0$. SYK computation involves taking fermion two-point function $\langle \overline{\psi}(\tau)\psi(0)\rangle$ dressing it with $\varphi$ as $\langle \overline{\psi}(\tau)\psi(0)e^{i\varphi(\tau)-i\varphi(0)}\rangle$ and then averaging over $\varphi$. The resulting expression is proportional to $e^{-\tau/K}$, hence we identify $K \propto 1/g_0$.

restrictive. *Importantly, the final result about the coupling bound will be applicable to any connectivity between different SYK.* More precisely, we consider two cases: each SYK is coupled to all others, and linear chain coupling[25]. Since we assumed that all the couplings are of the same order and there are no mixed correlators, the resulting large $N$ Schwinger–Dyson(SD) equations reduce to a single SYK SD equation with effective $J_{\text{eff}}$:

$$\text{all-to-all: } J_{\text{eff}}^2 = cJ^2, \quad \text{linear}: \ J_{\text{eff}}^2 = J^2. \tag{5.20}$$

Four-point function kernel is a $c$ by $c$ matrix. A straightforward analysis of this matrix reveals that the coefficient in front of Schwarzian(which is exactly the renormalized dilaton $\phi_r$) is $c$ times bigger than the $U(1)$ sigma-model coupling for *each* $U(1)$:

$$\text{both all-to-all and linear: } \phi_r = \alpha_S \frac{cN}{J_{\text{eff}}}, \quad \frac{1}{g_0} \sim K = \alpha_c \frac{N}{J_{\text{eff}}}, \tag{5.21}$$

where $\alpha_S \approx 0.0067, \alpha_c \approx 1.04$. These are the same numerical coefficients as in the complex SYK model. They can be tuneable if we allow extra disorder couplings or make the disorder couplings have different strengths. However, as we have mentioned above, this often makes the low energy physics deviate from JT. It would be interesting to explore how much one can deform $\alpha_S, \alpha_c$ while keeping JT physics. Returning to the actual expressions for $\phi_r, K$, one intuitive way to see that $\phi_r$ is $c$ times bigger is from the thermodynamics: both Schwarzian and the sigma-model contribute to free energy. But the sigma model involves only the corresponding SYK dot, whereas the reparametrizations involve the whole system.

Finally, let us look at the bounds. We need the equation (5.19), SYK UV cutoff $\epsilon_p \sim 1/J_{\text{eff}}$ and the fact that extremal entropy $S_0$ is additive when we couple SYK dots together. The bound (1.7) on $\phi_r$ can be rewritten as $N \lesssim e^{\mathcal{O}(1)N}$, which holds automatically for large $N$. Similarly, the bound on $g_0$ can be rewritten as $\log N \lesssim \frac{\pi}{2} e^{\mathcal{O}(1)N}$.

### 5.3.2 4d magnetic black holes

Near-extremal black holes in 4d develop a long, near-$AdS_2$ throat, with almost constant transverse $S^2$ radius. This closely mimics our eternal black hole setup, as the region outside the throat is close to being flat[26]. Similar to SYK, we need to somehow get matter CFT with central charge $c$ and $U(1)^c$ gauge group. One way to do it is to start from $c$ Dirac fermions in 4d, each charged under a separate $U(1)$. The black hole has a *magnetic* charge 1 with respect

---

[25]In this case it is possible to write down a more general $1 + 1$ effective action [72]. However, for our purposes we assume that all reparametrizations are homogeneous along the chain.

[26]One difficulty is that $S^2$ radius is not constant there, unlike our 2d setup.

to each $U(1)$. Recently, this setup has been explored in detail in the context of traversable wormholes [73–75]. One feature of magnetic black holes is the presence of fermionic zero modes(Callan–Rubakov effect [76, 77]). In the current setup with a unit magnetic charge, each 4d Dirac fermion gives rise to a single chiral 2d fermion. Since it is chiral, we should double the number of 4d fermions to include the ones with the opposite charge, such that they descend to 2d fermions with the opposite chirality. We assume that all $U(1)$ has the same coupling $g_{4d}$(in QFT it should be defined as the coupling at the scale of black hole radius). Then the renormalized dilaton value[27] $\phi_r$, the extremal entropy $S_0$ and 2d gauge coupling are(up to numerical coefficients):

$$\phi_r \propto \frac{r_e^3}{l_p^2}, \quad S_0 = \frac{\pi r_e^2}{l_p^2}, \quad g_0 \propto \frac{g_{4d}}{r_e}, \tag{5.22}$$

where $l_p$ is 4d Planck length. These equations are, in some sense "kinematic": they follow from having a near-extremal black hole with radius $r_e$. The actual dynamical input is the relation between $r_e$ and other parameters. In general it is adjustable:

$$r_e \geq \frac{\sqrt{\pi c} l_p}{g_{4d}}. \tag{5.23}$$

Exact equality corresponds to the situation when the black hole is not charged under any other gauge groups. Adding extra charges increases $r_e$. This inequality does not impose any restrictions on $g_{4d}$, rather it is a restriction on possible near-extremal black hole radii. In the presence of a large number of fields, a Bekenstein-like argument suggests actual UV cutoff is $\sqrt{c} l_p$ [78, 79] rather than $l_p$. When we use 4d parametrization (5.22) in our bounds, many factors nicely combine together. Starting from the bound on the dilaton (1.7), we can rewrite it in terms of one variable $x$:

$$x^3 \lesssim \exp(\mathcal{O}(1) x^2), \ x = \frac{r_e}{l_p \sqrt{c}} \gg 1. \tag{5.24}$$

Parameter $x$ is large because the extremal radius should be big in Planck units, otherwise the whole construction breaks down. Hence the bound is satisfied.

Similarly, our bound on $g_0$ implies

$$g_{4d} \gtrsim x \exp\left(-\frac{\pi}{2} e^{\mathcal{O}(1) x^2}\right), \tag{5.25}$$

---

[27]This can be read from near-extremal entropy $S(T) - S_0 \propto r_e^3 T / l_p^2$

whereas (5.23) says that

$$g_{4d} \gtrsim \frac{1}{x}. \tag{5.26}$$

For large $x$ the later one is more strict. Hence we conclude that for 2d gravity theories obtained from dimension reduction from 4d, our bound is automatically satisfied.

We see that at least in simple "bottom-up" constructions of JT theory with matter and gauge fields our bound is satisfied. This is a nice feature which demonstrates that the bound is physically reasonable. It would be very interesting to find examples where the bound is not automatic.

## 6    Conclusion

The main idea of this paper is that charge fluctuations can be used to bound entanglement entropy from below. Charge fluctuations in a region $R$ are very easy to extract from the expectation value of $\langle e^{i\alpha Q_R} \rangle$ and these calculations do not require replicas. Moreover it is expected that the measurement of simple operators like $e^{Q_R}$ should not be sensitive to islands. We explored a 2d setup and demonstrated that this lower bound on entropy generically grows as $c \log(\log(l))$ at zero temperature, where $l$ is the size of spatial region. This is inconsistent with the recent island computations for eternal black hole and bra-ket wormhole. Specifically, CFT fluctuation entropy can exceed black hole coarse grained entropy. Along the way, using precise predictions from the island formula, we to derived the bound (1.7) on the dilaton value in $JT$ gravity. This bound is not related to presence of gauge symmetries.

If the symmetry is global there is nothing which can save us from the paradox. This is not very surprising, as it has been conjectured a long time ago that full quantum gravity does not admit global symmetries. Obviously, something should change if the symmetry is gauged. We argued that in 2d nothing much really changes: 2d gauge bosons do not have local degrees of freedom, and at short distances gauge coupling can be ignored. What really saves us is the confining behavior of 2d gauge fields at long distances: there are no charge fluctuations if the charges are screened. This comes with a price of imposing a lower bound on the gauge coupling. It is quite obvious that if the gauge fields are everywhere and once the coupling is big, the charge fluctuations will be suppressed. What is not trivial, is that we demonstrated that the same happens if the gauge fields are included in the gravitational region *only*. This suggests that it is indeed the feature of gravity(and not of our particular setup) to impose the bound (1.6). We checked the above bound in some "bottom-up" models of JT, such as SYK and 4d near-extremal magnetic black holes, and saw that the bound is indeed satisfied. Also we argued that it is difficult to get rid of large fluctuation entropy by

including small corrections(even non-perturbative ones) to the computation of $\langle e^{i\alpha Q_R} \rangle$. This bound obviously has a flavor of weak gravity conjecture. In higher dimensions one typically imposes a constraint on mass to charge ratio. In our 2d setup we studied CFT (which is obviously massless) coupled to JT gravity. What we got is that the gauge coupling can not be much smaller than the dilaton value.

Obviously, it is important to understand what happens in higher dimensions, especially 1+3. Here we do not expect all gauge fields to be confining, even at large distances. One such example is QED. The charge distribution again can be found from computing the expectation value of $e^{i\alpha Q}$, which again does not involve replicas. One possible resolution in higher dimensions might be related to the fact that Hawking radiation is dominated by low angular momentum modes. So that effectively the physics is always 2d. Although it might be challenging to perform this computation precisely as the transverse two-sphere far away from the horizon does not have a constant radius, so one cannot perform a naive reduction to 2d. It would be very interesting to do this computation.

One can wonder what are other possible effects which can lower fluctuation entropy. We argued that small non-perturbative corrections of order $e^{-S_0}$ in the computation of $\langle e^{i\alpha Q_R} \rangle$ cannot decrease fluctuation entropy much. In principle, corrections of order $e^{-S_0/c}$ corresponding to "fractional" baby universes can do the job, but we are not aware of any proposals in the literature of why they should exist.

In this paper we mostly concentrated on massless matter. If a small mass is present, we expect that it will damp the fluctuations. Hence, larger values of the mass allow smaller values of the coupling. This contrasts with the weak gravity conjecture, which roughly says that the minimal coupling grows with the mass, $g_{4d} \gtrsim l_p m$.

Very recently ref. [80] has raised some concerns regarding the island prescription and gravitational Gauss law, as the energy inside the island can be measured at infinity outside the radiation region. Without diving into the details, let us mention that ref. [80] proposes that the graviton mass would resolve this tension. Obviously, the same concerns can be expressed regarding electric charge. However, in this paper the island prescription is not important for our results and moreover we studied in some sense a "complimentary" quantity, the fluctuation entropy, which measures charge fluctuations. Interestingly, we also found that possible tensions with gravity can be resolved if the gauge boson is massive. In our case gauge boson automatically acquires mass due to the Schwinger effect and we do not need to add it by hand. It would be interesting to explore this similarity further.

## Acknowledgment

We would like to thank Yiming Chen, Xi Dong, Alexander Gorsky, Thomas Hartman, Luca Iliesiu, Zohar Komargodski, Sean McBride, Juan Maldacena, Donald Marolf, Henry Maxfield, Mark Mezei, Geoff Penington, Fedor Popov, Andrey Sadofyev, Nikolay Sukhov, Wayne Weng and Zhenbin Yang for comments and discussions. AM would like to thank Cory King for moral support. The work of AM was supported by the Air Force Office of Scientific Research under award number FA9550-19-1-0360. The work of AM was also supported in part by funds from the University of California. The work of AT was supported in part by a grant from the Simons foundation and in part by funds from the University of California.

## A    Variance sets maximal entropy

In this paper we studied coupling to electromagnetic forces, but it is interesting to ask what happens in interacting CFTs. Now we argue that generic interactions do indeed lower the fluctuation entropy. Thus, in principle, they are capable of resolving the paradox too. Or, at least, the presence of other interactions can allow a smaller value of the gauge coupling. For simplicity, we present the argument for the case of $U(1)$. The central point is that conformal symmetry fixes the charge variance in a given interval, eq. (2.29):

$$\langle Q^2 \rangle_{2d\ CFT} = \frac{k}{\pi^2} \log \frac{l}{\epsilon_{\text{uv}}} \tag{A.1}$$

If the currents admit a free field representation(as is the case for free fermions or, more generally, Luttinger liquid) one can compute full charge distribution $p(q)$, eq. (2.21), and the fluctuation entropy (2.22). If the currents are not free, the variance $\langle Q^2 \rangle$ in eq. (A.1) imposes an upper bound on the fluctuation entropy. This upper bound is saturated by free currents. This statement is a simple exercise in classical probability theory. We are looking for a classical probability distribution $p(q)$ which maximises the Shannon entropy, but has a fixed second moment $\langle Q^2 \rangle$. We can immediately write down a variational problem with Lagrange multipliers $\lambda, \eta$:

$$-\sum_q p(q) \log p(q) + \lambda \left( \sum_q p(q) - 1 \right) + \eta \left( \sum_q q^2 p(q) - \langle Q^2 \rangle \right). \tag{A.2}$$

The solution is a Gaussian distribution for $p(q) \propto e^{-\#q^2}$, which corresponds to free current result (2.21). The corresponding entropy in the limit of large $\langle Q^2 \rangle$:

$$S_{f,\mathrm{max}} = \frac{1}{2} \log \left( 2\pi \langle Q^2 \rangle \right) + \frac{1}{2}. \tag{A.3}$$

This last equality holds in any number of dimensions. This is the maximum[28] of the entropy. Heuristic way to see this is by noticing that the minimum entropy can be achieved by forcing $p(q)$ to be supported on only two values of $q$, saturating the variance. Slightly more complicated distributions can even achieve zero entropy. The upshot is that the value of the charge variance puts an upper bound on the fluctuation entropy. In 2d CFTs this upper bound is saturated for free currents.

Interestingly, *if* $\log p(q)$ *is a concave function(i.e. $p(q)$ is log-concave)*, there is an lower bound on the entropy from variance [81, 82]:

$$S_{f,min} = \frac{1}{2} \log \left( 4 \langle Q^2 \rangle \right). \tag{A.4}$$

Hence, curiously, if we want to avoid large fluctuation entropy the charge distribution has to be a log-concave function.

# B    Fluctuation entropy for $U(N)$

In this Appendix we compute the probability distribution $p(\lambda)$ for level $k$ $U(N)$ WZW model:

$$p(\lambda) \propto \dim(\lambda) \int d\alpha \; \mu_{\mathrm{Haar}}(\alpha) \chi_\lambda(\alpha) \exp \left( -\frac{1}{2g} \sum_a \alpha_a^2 \right). \tag{B.1}$$

"Coupling constant" $g$ is determined by the two-point function of the bosonized field $\phi$. At finite temperature

$$g = \frac{2\pi^2}{k \log \left( \frac{\beta}{\pi \epsilon_{\mathrm{uv}}} \sinh \left( \frac{\pi l}{\beta} \right) \right)}. \tag{B.2}$$

Normalization constant can be found from

$$\sum_\lambda p(\lambda) = 1. \tag{B.3}$$

---

[28] There is also a zero-mode which is the mean value of the charge. It does not affect the value of the entropy, and we assume it to be zero.

We will be interested in the regime $g \ll 1$, $gN \ll 1$. The last inequality is stronger than the first one if $N$ is allowed to be large. The calculation is very similar to evaluating Wilson loop expectation value in Gross–Witten–Wadia matrix model at extremely weak coupling(hence we are far away from the phase transition), we refer to [83] for recent discussion.

Both Haar measure and representation $\lambda$ character(which is just a Schur polynomial for the case of $U(N)$) can be written as a determinant:

$$\Delta(e^{i\alpha}) = \prod_{a<b} \left( e^{i\alpha_a} - e^{i\alpha_b} \right), \tag{B.4}$$

$$\mu_{\text{Haar}}(\alpha) = |\Delta(e^{i\alpha})|^2, \tag{B.5}$$

$$\chi_\lambda(\alpha) = \frac{1}{\Delta(e^{i\alpha})} \det_{a,b} \exp\left( i\alpha_a(\lambda_b + N - b) \right), \tag{B.6}$$

where we have parametrized the representation $\lambda$ with the corresponding Young diagram.

The determinants in eq. (B.1) can be expanded in terms of sum over two permutations $\rho, \sigma ((-1)^{|\rho|,|\sigma|}$ is the signature):

$$\int d\alpha \ \exp\left( -\frac{1}{2g} \sum_a \alpha_a^2 \right) \sum_{\sigma,\rho} (-1)^{|\sigma|+|\rho|} \prod_a \exp\left( -i(N-a)\alpha_{\rho_a} + i(N-a+\lambda_a)\alpha_{\sigma_a} \right). \tag{B.7}$$

The integrand is permutation invariant. So we can actually put $\sigma_a = a$:

$$\int d\alpha \ \exp\left( -\frac{1}{2g} \sum_a \alpha_a^2 \right) \sum_\rho (-1)^{|\rho|} \prod_a \exp\left( i\alpha_a(\lambda_a - a + \rho_a) \right) =$$

$$= \text{const} \, g^{N/2} \exp\left( -\frac{g}{2} \sum_a (\lambda_a - a)^2 \right) \sum_\rho (-1)^{|\rho|} \prod_a \exp\left( -g\rho_a(\lambda_a - a) \right). \tag{B.8}$$

In the above we exploited the fact that $g \ll 1$, hence the integral over $\alpha$ can be extended to $(-\infty, +\infty)$. The sum over $\rho$ is equal to Vandermonde determinant times the Schur polynomial:

$$\Delta\chi_\lambda, \tag{B.9}$$

both of which are evaluated at the set of variables $(e^{-g}, e^{-2g}, \dots)$. The Vandermode simply gives $g^{N(N-1)/2}$. Even if we assume $gN \ll 1$, Schur polynomial is not equal to just $\dim(\lambda)$. The proper expression in the regime $gN \ll 1$ is ( [84], paragraph I3, example 1):

$$\chi_\lambda(e^{-g}, e^{-2g}, \dots) = \exp\left( -g \sum_a a\lambda_a \right) \dim(\lambda). \tag{B.10}$$

Putting everything together we get($|\lambda|$ is the number of boxes):

$$p(\lambda) = Z^{-1} \dim(\lambda)^2 \exp\left( -\frac{g}{2} C_2(\lambda) + g|\lambda|(N+1)/2 - g\sum_a a\lambda_a \right), \qquad \text{(B.11)}$$

where $Z$ is normalization constant:

$$Z = \sum_\lambda \dim(\lambda)^2 e^{-\frac{g}{2}C_2(\lambda)+g|\lambda|(N+1)/2-g\sum_a a\lambda_a}, \qquad \text{(B.12)}$$

and $C_2(\lambda)$ is the quadratic Casimir:

$$C_2(\lambda) = \sum_a \lambda_a(\lambda_a - 2a + N + 1). \qquad \text{(B.13)}$$

To reiterate, the result (B.11) holds as long as $gN \ll 1$. If $N$ is finite then, due to $\dim(\lambda)$, the probability distribution is localized at $\lambda_a \sim 1/\sqrt{g}$. It means that we can neglect all the terms in the exponent except $C_2(\lambda)$. This way we recover the $SU(2)$ result of [5] and the recent results of [37]. In the limit $N \gg 1$ we expect that the sum over $\lambda$ is actually dominated by a saddle point. Interestingly, $Z$ looks very similar to 2d YM sphere partition function. Precise identification is spoiled by the $\sum_a a\lambda_a$ term and it also makes the computations difficult. We can assume that this term and $N|\lambda|$ are actually small compared to $C_2(\lambda)$. We will verify this assumption a posteriori: we will see that the sum is dominated by a Young diagram with row lengths $\lambda_a \sim N/\sqrt{Ng}$ and hence this assumption holds as long as $Ng \ll 1$. This expression also agrees with the recent results of [37] obtained by different means.

We will find the fluctuation entropy only in the regime $N \gg 1, Ng \ll 1$. It would be convenient to introduce

$$Z_\varkappa = \sum_\lambda \dim(\lambda)^\varkappa e^{-\frac{g}{2}C_2(\lambda)}, \qquad \text{(B.14)}$$

which is exactly the partition function of 2d YM on a surface with Euler characteristic $\varkappa$. Since the coupling $Ng$ is very small we are far away from Kazakov–Douglas [85] phase transition. Representation dimension in the large $N$ limit also has a nice expression in terms of $\lambda_a$:

$$\dim(\lambda) = \prod_{a>b}\left(1 - \frac{\lambda_a - \lambda_b}{a-b}\right) \qquad \text{(B.15)}$$

Without going into the details, we simply state the answer for $Z_\varkappa$, [85]:

$$Z_\varkappa = \exp\left( -\frac{N^2 \varkappa}{4} \log \frac{2Ng}{\varkappa}(1 + \mathcal{O}(1/N)) + \frac{N^3 g}{24} \right). \qquad \text{(B.16)}$$

Moreover, the corresponding saddle point Young diagram has row length $\lambda_a \sim N/\sqrt{Ng}$. This justifies our assumption.

Finally, we are in position to evaluate the fluctuation entropy:

$$S_f = -\sum_\lambda p(\lambda) \log \frac{p(\lambda)}{\dim(\lambda)}. \tag{B.17}$$

It is straightforward to check that in terms of $Z_\varkappa$ it is given by:

$$S_f = -g\frac{\partial \log Z_2}{\partial g} - \frac{\partial \log Z_\varkappa}{\partial \varkappa}\bigg|_{\varkappa=2} + \log Z_2. \tag{B.18}$$

Evaluating this expression yields the following $\log l$ dependence:

$$S_f = \frac{N^2}{4} \log \left( k \log \left( \frac{\beta}{\pi \epsilon_{\mathrm{uv}}} \sinh \left( \frac{\pi l}{\beta} \right) \right) \right). \tag{B.19}$$

One thing to notice is that only the expectation value of $\log(\dim(\lambda))$ contributes, as the probability distribution is dominated by a single representation.

# C   Bounding the entropy

In this Appendix we would like to obtain bounds on $\rho_{gauge}^{semi}(R \cup I)$. We can use Araki–Lieb inequality: for any two subsystems $A, B$ the following inequality holds

$$|S(\rho(A)) - S(\rho(B))| \leq S(\rho(A \cup B)), \tag{C.1}$$

which implies:

$$S(\rho(A)) \leq S(\rho(B)) + S(\rho(A \cup B)). \tag{C.2}$$

In our case we put $A = R \cup I$ and $B$ is the rest, $B = \overline{R \cup I}$. One can easily see that $\rho_{gauge}^{semi}(R \cup I)$ in eq. (4.18) descends from the following mixed state:

$$\rho_{gauge}^{semi}(R \cup I) = \mathrm{Tr}_{\overline{R \cup I}} \, \rho_{gauge}^{semi}(\mathrm{vac}), \tag{C.3}$$

where

$$\rho_{gauge}^{semi}(\mathrm{vac}) = \int_{-\pi}^{\pi} d\gamma \, e^{iQ_I\gamma}|\mathrm{vac}\rangle\langle\mathrm{vac}|e^{-iQ_I\gamma}, \tag{C.4}$$

and

$$\mathrm{Tr}_{R \cup I} \, \rho_{gauge}^{semi}(\mathrm{vac}) = \rho^{semi}(\overline{R \cup I}). \tag{C.5}$$

Moreover, a straightforward computation shows that in both Abelian and non-Abelian cases[29] the entropy of $\rho_{gauge}^{semi}(\text{vac})$ is equal to the fluctuation entropy:

$$S(\rho_{gauge}^{semi}(\text{vac})) = S_f(I). \tag{C.7}$$

Therefore,

$$S(\rho_{gauge}^{semi}(R \cup I)) \leq S(\rho^{semi}(R \cup I)) + \min(S_f(I), S_f(R)). \tag{C.8}$$

The minimum arises because in defining $\rho_{gauge}^{semi}(R \cup I)$ we can use either $Q_R$ or $Q_I$.

We can obtain another lower bound by noticing that $\rho_{gauge}^{semi}$ commutes with $Q_R$ and $Q_I$. Hence we can again bound the full entanglement entropy with the fluctuation entropy:

$$\max\left(S_f(I), S_f(R)\right) \leq S\left(\rho_{gauge}^{semi}(R \cup I)\right). \tag{C.9}$$

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
