# Peer review of "Charge fluctuation entropy of Hawking radiation: a replica-free way to find large entropy"

_SciPost Physics_

## Round 2 · Referee Report · Anonymous (Referee 1) · 2022-12-20

Report

Dear Editor,

In this work, the authors study a fixed charge version of matter entanglement entropy, and use the information paradox argument to argue that it should be lower than the full black hole entropy, and computable without using replica trick techniques.
This then leads to paradoxes: for a global symmetry, the paradox is unresolved and agrees with the absence of global symmetries in quantum gravity. For a gauged symmetry,
the paradox is avoided if the gauge coupling is sufficiently large, for which an estimate is obtained.
The paper is timely, well-written, and implements an interesting idea in full detail. It contains careful and detailed calculations, supplemented with physically motivated estimates.
A lot of attention is paid to assessing the range of validity of the proposed bound by looking at non-perturbative corrections, matter interactions, relation to SYK and magnetic black holes...

As such, I find the paper immediately suitable for publication.

Perhaps one small comment on notation: equation (2.1) would be clearer with a direct sum symbol.

---

## Round 2 · Referee Report · Henry Maxfield (Referee 2) · 2023-1-13

Report

The manuscript provides a novel and interesting connection between different topics: entropy in the presence of symmetries and entropy of black holes. The results are clearly explained with sufficiently detailed arguments, and usefully summarised in the abstract and introduction. I recommend publication once a small clarification is addressed in part of section 2.3.

The notation is somewhat unclear and confusing from (2.33) to (2.37), so some change or clarification would be helpful. $\alpha$ is specified to be an element of the Lie algebra, but then it is not clear what $Q_R$ in (2.33) means precisely. Presumably $e^{i\alpha Q_R}$ means the operator representing the group element $e^{i\alpha}$ on Hilbert space, so perhaps $Q_R$ is a coadjoint-valued operator and $\alpha Q_R$ includes the pairing between adjoint and coadjoint. After that, the terminology for the Haar measure is slightly confusing: ordinarily this would refer to the measure $dU$ on the group, but above (2.36) it is used instead for the measure after reducing to the eigenvalues $\alpha_a$.

Some other minor comments in case the authors find them useful:

Regarding section 5.2, it may be possible to use a two-replica calculation to estimate the ensemble variance of $\langle e^{iQ_R\alpha}\rangle$. If this becomes as large at late times as the square of the mean value computed in the paper, it would indicate that the fluctuation entropy one-replica calculation may be unreliable. While this would be helpful to consider if possible, it is certainly not necessary for publication.

In section 4.2 I worried for a while that zero temperature may be problematic since the Schwarzian mode becomes strongly coupled at low temperature. This is not in fact a problem because one can always work at a temperature large enough for weak coupling but with $b \ll \beta$, but it might be helpful to clarify this.

A very minor comment: before (2.5), the text specifies computing moments of $p(q)$, but throughout the characteristic function $\langle e^{iQ_R\alpha}\rangle$ is computed instead (and this is better, since in general the moments may not uniquely specify a distribution, while the characteristic function does).

---

## Editorial Decision

resubmitted